# Computational pathology identifies immune-mediated collagen disruption to predict clinical outcomes in gynecologic malignancies

Arpit Aggarwal [1,2], Sirvan Khalighi[2], Deepak Babu[3], Haojia Li [3], Sepideh Azarianpour-Esfahani [3], Germán Corredor [2,4], Pingfu Fu [3], Mojgan Mokhtari[5], Tilak Pathak[2], Elizabeth Thayer [6], Susan Modesitt[6], Haider Mahdi[7], Stefanie Avril[3] & Anant Madabhushi [1,2,8 ✉]

**Abstract**

**Background** The role of immune cells in collagen degradation within the tumor microenvironment (TME) is unclear. Immune cells, particularly tumor-infiltrating lymphocytes (TILs), are known to alter the extracellular matrix, affecting cancer progression and patient survival. However, the quantitative evaluation of the immune modulatory impact on collagen architecture within the TME remains limited.

**Methods** We introduce CollaTIL, a computational pathology method that quantitatively characterizes the immune-collagen relationship within the TME of gynecologic cancers, including high-grade serous ovarian (HGSOC), cervical squamous cell carcinoma (CSCC), and endometrial carcinomas. CollaTIL aims to investigate immune modulatory impact on collagen architecture within the TME, aiming to uncover the interplay between the immune system and tumor progression.

**Results** We observe that an increased immune infiltrate is associated with chaotic collagen architecture and higher entropy, while immune sparse TME exhibits ordered collagen and lower entropy. Importantly, CollaTIL-associated features that stratify disease risk are linked with gene signatures corresponding to TCA-Cycle in CSCC, and amino acid metabolism, and macrophages in HGSOC.

**Conclusions** CollaTIL uncovers a relationship between immune infiltration and collagen structure in the TME of gynecologic cancers. Integrating CollaTIL with genomic analysis offers promising opportunities for future therapeutic strategies and enhanced prognostic assessments in gynecologic oncology.

**Plain Language Summary**

The role of cells that are part of our immune system in altering the structure of the protein collagen within cancers is not fully understood, particularly within cancers that affect women such as ovarian, cervical and uterine cancers. Here, we developed a computer-based method called CollaTIL to explore how immune cells influence collagen in these tumors and affect their growth. We found that a higher presence of immune cells leads to less organized collagen in the tumor. Conversely, when there are fewer immune cells, the collagen tends to be more structured. Additionally, CollaTIL identifies patterns that predict patient outcomes in these cancers. These findings not only enhance our understanding of tumor biology but also may be useful in helping clinicians to predict which patients are at risk of their disease progressing.

[1] Georgia Tech, Georgia, GA, USA. [2] Emory University, Georgia, GA, USA. [3] Case Western Reserve University, Ohio, OH, USA. [4] Louis Stokes Cleveland Veterans Administration Medical Center, Ohio, OH, USA. [5] Isfahan University of Medical Sciences, Isfahan, Iran. [6] Emory University School of Medicine, Georgia, GA, USA. [7] University of Pittsburgh Medical Center, Pittsburgh, PA, USA. [8] Atlanta Veterans Administration Medical Center, Georgia, GA, USA. ✉email: anantm@emory.edu

The tumor microenvironment (TME) represents a multi-faceted ecosystem comprising tumor cells, stromal cells, immune cells, blood vessels, the extracellular matrix (ECM), and signaling molecules[1]. Each constituent of the TME exerts a profound influence on one another, intricately orchestrating the dynamics of cancer progression and therapeutic response[2]. Collagen, as the predominant protein in the ECM, assumes a pivotal role in cancer progression and metastasis[3,4]. Immune cells, particularly tumor-infiltrating lymphocytes (TILs), are key participants in the body's defense against cancer[5,6]. Recent investigations have shed light on the role of TILs in remodeling the ECM through enzymatic activity, inducing changes in its physical properties[7–10]. This ECM remodeling facilitates immune cell infiltration within the tumor, hinders tumor growth, and ultimately contributes to prolonged patient survival. Consequently, a pressing question arises: Does the augmented immune cell response correlate with heightened enzymatic activity, resulting in an increased breakdown of collagen architecture and the emergence of disordered collagen fibers?

Machine learning strategies have emerged as powerful tools for quantifying the immune milieu from Hematoxylin & Eosin (H&E) Whole Slide Images (WSIs), demonstrating their potential in prognostic assessment[6,11,12]. However, limited studies have explored the quantitative evaluation of the immune modulatory impact on collagen architecture within the TME. Previous findings suggest that an immune-rich TME exhibits increased enzymatic activity, leading to the enhanced breakdown of collagen architecture and the formation of more disordered collagen fibers[7–10]. Conversely, an immune-sparse TME is expected to have reduced enzymatic activity, resulting in the better preservation of collagen architecture.

Leveraging recent advancements in computational and machine learning tools, we introduce CollaTIL, a computational pathology method designed to quantitatively characterize the architecture and arrangement of immune and collagen components within the TME from H&E WSIs. Our primary objective with CollaTIL is to investigate the immune modulatory impact on collagen architecture within the TME and uncover the interplay between the immune system and tumor progression. This study focuses on gynecologic cancers, which are known to have immune-rich TME[11]. With CollaTIL, we aim to investigate the immune modulatory impact on collagen architecture within the TME by quantitatively characterizing the anti-tumor impact of the immune milieu through the disruption of collagen architecture. Specifically, our study employs CollaTIL to accomplish two key objectives. Firstly, we aim to quantitatively characterize and subsequently investigate the relationship between immune and collagen architecture within the TME of gynecologic cancers, including high-grade serous ovarian carcinoma (HGSOC), cervical squamous cell carcinoma (CSCC), and endometrial carcinoma (EC), along with evaluating the association of CollaTIL features with survival outcomes. Then, we endeavor to assess whether the morphological relationship between immune and collagen architecture, as assessed by CollaTIL, extends to the molecular level. By integrating genomic analysis, our aim is to identify relevant biological pathways, gene signatures, and unravel the distinct molecular dynamics associated with both favorable and unfavorable clinical outcomes in patients with HGSOC and CSCC.

Our study enables the extraction of features that capture the complex relationship between immune and collagen architecture within the TME. We find that heightened immune infiltrate within the TME correlates with chaotic collagen architecture and increased entropy, while immune-sparse TME exhibits ordered collagen with lower entropy. Furthermore, CollaTIL-derived features associated with disease risk stratification correspond to specific gene signatures, highlighting Tricarboxylic Acid (TCA) Cycle in CSCC and amino acid metabolism, and macrophages in HGSOC.

## Methods

**Patient populations for the study.** We analyzed 493 patients diagnosed with HGSOC ($n = 139$), CSCC ($n = 269$), or EC ($n = 85$) from various sites, including The Cancer Genome Atlas (TCGA, $n = 357$)[13], University Hospitals (UH, $n = 58$), Cleveland Clinic (CCF, $n = 48$), and Memorial Sloan Kettering Cancer Center (MSKCC, $n = 30$)[14]. For the patients from TCGA diagnosed with HGSOC (D0, $n = 95$), tissue samples were obtained through cytoreductive surgery before the initiation of chemotherapy. Similarly, for the patients from TCGA diagnosed with CSCC (D1 and D2, $n = 262$), tissue samples were obtained through cytoreductive surgery before the initiation of chemotherapy or radiotherapy. All patients from UH (D3 and D4, $n = 58$) underwent surgery (hysterectomy with bilateral salpingo-oophorectomy), the standard treatment for EC. The patients in these cohorts who were considered as intermediate- or high-risk for recurrence received chemotherapy following surgery (D3, $n = 32$). The CCF cohorts (D5, D6, and D7) included 48 patients treated with immunotherapy agents, including pembrolizumab, nivolumab and/or ipilimumab, and avelumab, all in the recurrent setting (D5 for HGSOC, D6 for CSCC, and D7 for EC). All patients from MSKCC (D8, $n = 30$) were diagnosed with HGSOC and underwent primary debulking surgery. The training cohort (D0) consisted of 95 patients from TCGA diagnosed with HGSOC, while the validation cohorts (D1-D8) comprised a total of 398 patients diagnosed with HGSOC, CSCC, or EC from various sites. The cohorts (D0-D8) comprised 71 patients with stage IV disease, 153 with stage III, 68 with stage II, 192 with stage I, and 9 with an unknown stage.

Patients from TCGA[13], UH, CCF, and MSKCC[14] cohorts underwent a standardized process for tissue preparation, which involved formalin fixation and paraffin-embedding, followed by H&E staining. The resulting tissue sections were digitized as WSIs at 40x magnification with a resolution of 0.25 microns per pixel. Prior to analysis, the WSIs underwent rigorous quality control checks to exclude any images with artifacts or blurriness, utilizing the HistoQC tool[15]. We included patients from these sites (TCGA, UH, CCF, and MSKCC) that had at least one H&E WSI, and their overall survival (OS) or progression-free survival (PFS) outcome information was available. The PFS outcome information was available for patients from CCF and MSKCC (D5-D8), while the molecular data, including mutation annotation, and genes expressions were available for patients from TCGA (D0-D2). All H&E imaging was carried out before the initiation of therapy (Fig. 1, Supplementary Table 1, and Supplementary Methods).

**CollaTIL framework.** The CollaTIL framework comprises of multiple image analysis routines for characterizing the immune and collagen architecture on H&E WSIs (Fig. 2 and Supplementary Tables 2–3). In CollaTIL module a, we implemented preprocessing steps, such as epithelium/stroma and nuclei segmentation, to the H&E WSIs. Specifically, we extracted a set of non-overlapping 3000×3000-pixel tiles from each H&E WSI. We extracted multiple non-overlapping 3000 × 3000-pixel tiles from the H&E WSI because processing these enormous H&E WSIs in their entirety would be unrealistic due to computational limitations. An existing pretrained deep learning model was used to segment the epithelial and stromal regions on these tiles[16]. The Hovernet model[17], a state-of-the-art method for nuclei

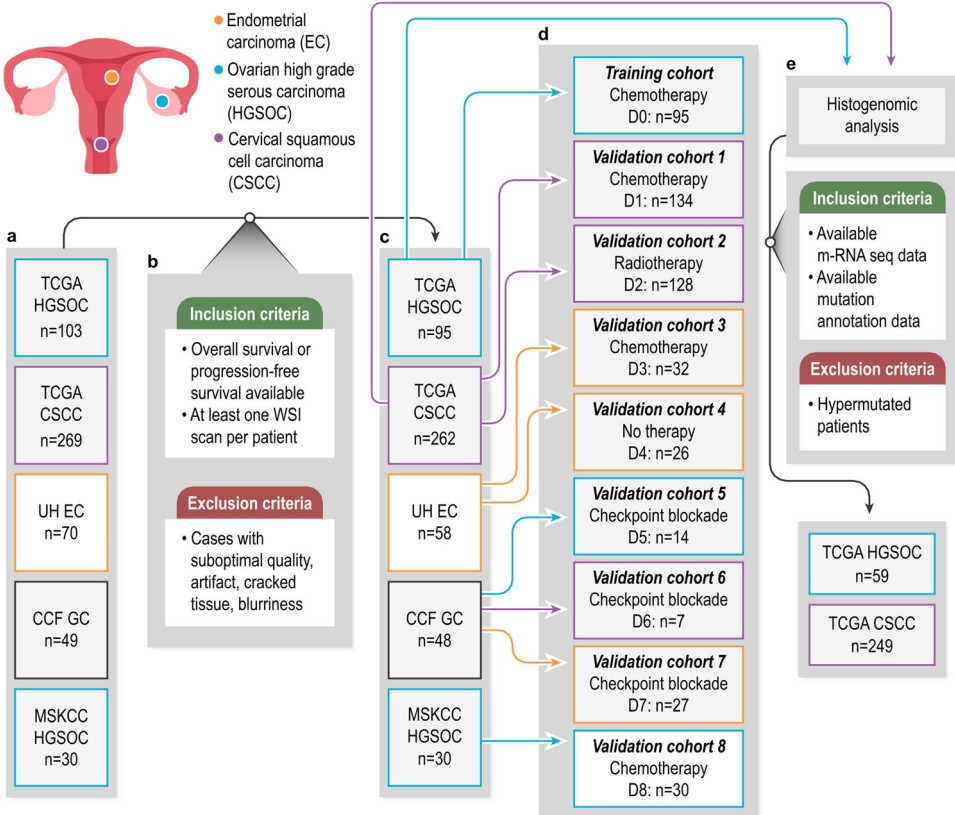

**Fig. 1 Patient selection diagram for the cohorts included in this study. a** The number of patients available for each cancer type (HGSOC, CSCC, and EC) from various sites (TCGA, UH, CCF, and MSKCC). **b** The inclusion and exclusion criteria followed for this study. Only patients with available overall survival or progression-free survival information and at least one H&E WSI were included. **c** The number of patients available for each cancer type (HGSOC, CSCC, and EC) from various sites (TCGA, UH, CCF, and MSKCC) after following the inclusion and exclusion criteria. **d** The training cohort and validation cohorts used for the study. The training cohort (D0) consisted of 95 patients with high grade serous ovarian carcinoma from the TCGA site. There were eight validation cohorts (D1-D8) used for the study. **e** The inclusion and exclusion criteria followed for performing the histogenomic analysis. This analysis was performed on two cohorts, one consisting of 59 patients with high grade serous ovarian carcinoma from the TCGA site and the other consisting of 249 patients with cervical squamous cell carcinoma from the TCGA site. TCGA, The Cancer Genome Atlas; UH University Hospitals, CCF Cleveland Clinic, MSKCC Memorial Sloan Kettering Cancer Center, WSI whole slide image, GC gynecologic cancer, EC Endometrial carcinoma, HGSOC Ovarian high grade serous carcinoma, CSCC Cervical squamous cell carcinoma, OS overall survival, PFS progression-free survival.

segmentation and classification, was then applied for the segmentation of nuclei on these tiles. To evaluate the performance of these models, we conducted a visual assessment with two pathologists. They independently reviewed one randomly selected 3000 × 3000-pixel tile from 50 different patients randomly chosen from the training cohort (D0). The two pathologists examined the tiles and categorized them into one of three categories (good, fair, or poor). For the nuclei segmentation, both pathologists unanimously ranked 100% of the tiles as either good or fair. However, for the epithelium/stroma segmentation, the first pathologist ranked 90% of the tiles as good or fair, while the second pathologist assigned a good or fair ranking to 94% of the tiles (Fig. 2a, Supplementary Methods, and Supplementary Table 4).

CollaTIL module b illustrates the process of extraction of features from the collagen component within the TME. Each tile of the H&E WSI was divided into an array of tumor neighborhoods. Following the segmentation of stromal regions, a derivative-of-Gaussian based model was utilized to capture the fiber orientations by detecting linear structures within these regions. Within each tumor neighborhood, the orientation co-occurrence matrix was constructed based on the set of fiber orientations. The quantitative measurement of collagen fiber orientation disorder in stromal regions (CFOD-S) was calculated from this matrix based on entropy theory, where entropy

represents the level of uncertainty in the collagen fiber's orientation within the tumor neighborhood. To evaluate the performance of the model used for collagen fiber segmentation, we conducted a visual assessment with two pathologists, as previously described. The first pathologist ranked 90% of the tiles as good or fair, while the second pathologist assigned a good or fair ranking to 92% of the tiles (Fig. 2b, Supplementary Methods, and Supplementary Tables 2–4).

CollaTIL module c illustrates the process of extraction of features from the immune component within the TME. This was achieved by classifying the segmented nuclei in each tile as either TIL or non-TIL using a pre-existing machine learning model[18]. Based on the previously segmented epithelial and stromal regions, four different classes of nuclei were defined: epithelial TILs, epithelial non-TILs, stromal TILs, and stromal non-TILs. The density of each nuclei class was calculated. Additionally, the architecture and interplay of each nuclei class were characterized by constructing clusters based on proximity. To evaluate the performance of the model used for TIL detection, we conducted a visual assessment with two pathologists, as previously described. The first pathologist ranked 90% of the tiles as good or fair, while the second pathologist assigned a good or fair ranking to 94% of the tiles (Fig. 2c, Supplementary Methods, and Supplementary Tables 2–4).

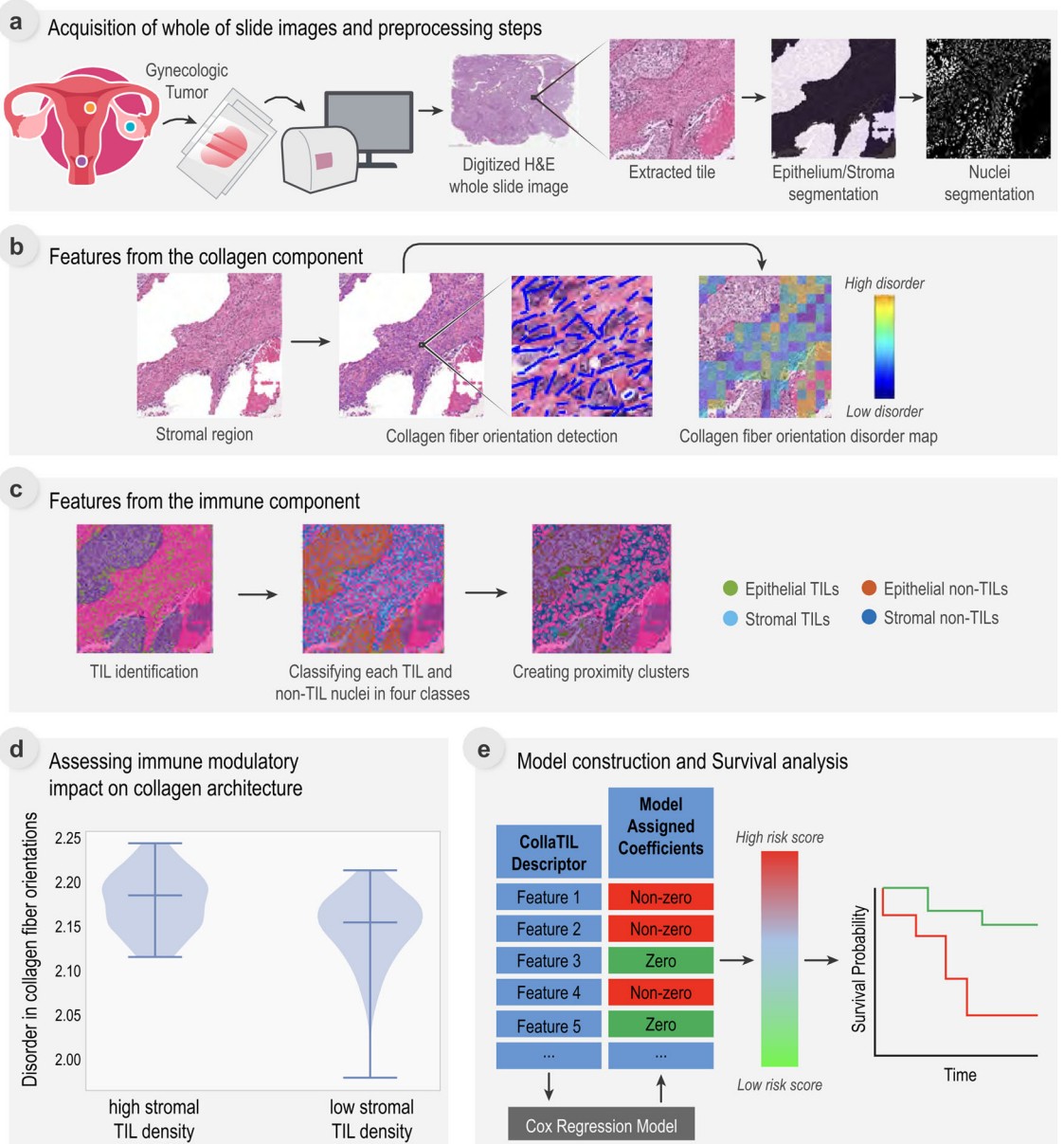

**Fig. 2 Overall framework of CollaTIL. a** H&E WSIs were collected from TCGA, UH, CCF, and MSKCC sites. For each H&E WSI, tiles of dimension 3000 × 3000 were extracted. The preprocessing steps required for feature extraction, epithelium/stroma segmentation and nuclei segmentation were performed on each extracted tile from the H&E WSI. **b** First set of features related to quantitative features of Collagen Fiber Orientation Disorder in Stromal regions were extracted. The collagen fiber orientations in stromal regions were captured using a derivative-of-Gaussian based model. An orientation co-occurrence matrix was constructed with a brighter co-occurrence value in the on-diagonal cells suggesting higher co-occurrence of collagen fibers of the same orientation. The feature quantifying the degree of disorder of collagen fiber orientations was then calculated from this matrix. **c** Another set of features related to quantitative characterization of architecture of tumor infiltrating lymphocytes (TILs) and their interplay with surrounding cells were extracted. **d** Assessing immune modulatory impact on collagen architecture by comparing the TIL density and disorder in collagen fiber orientations in the high and low-risk patients identified by CollaTIL. **e** CollaTIL consisted of features obtained from **b** and **c**. A Cox Regression Model was constructed on D0 cohort to discover top features by assigning a corresponding coefficient to each of the features, based on which a continuous risk score was generated. Kaplan–Meier survival analysis was performed on training and validation cohorts. TCGA, The Cancer Genome Atlas, UH University Hospitals, CCF Cleveland Clinic, MSKCC Memorial Sloan Kettering Cancer Center, TIL Tumor-infiltrating lymphocytes, H&E WSI Hematoxylin and Eosin Whole Slide Image, OS overall survival, PFS progression-free survival.

A Cox proportional hazard model[19] (referred to as Cox regression model) with elastic net penalty[20] was trained using the CollaTIL features on the D0 cohort to predict OS. A coefficient was assigned to each feature in the final model and a continuous risk score was obtained by linear combination of top features weighted by corresponding coefficients for each patient. The continuous risk score for each patient reflects an estimated risk for OS or PFS and was converted to a binary high vs low value using the mean threshold. We validated CollaTIL on the D1-D4 cohorts for predicting OS and the D5-D8 cohorts for predicting PFS, using the same set of feature coefficients. Kaplan–Meier survival analysis with the log-rank test was used to examine the differences of time-to-event data between the two patient groups. The performance of models was summarized by hazard ratios

(HRs) along with their 95% confidence intervals (CIs) using Wald test and Harrell's concordance index (c-index) on the D1-D8 validation cohorts. Statistical significance was determined using a two-sided threshold of $p < 0.05$ (Fig. 2d, e). Univariate and multivariable analyses were performed to assess the prognostic significance of clinicopathological variables (Age, Tumor grade, International federation of gynecology and obstetrics (FIGO) stage, Molecular subtypes, and Human papillomavirus (HPV) dependency) and the CollaTIL signature on OS/PFS (Supplementary Methods and Supplementary Table 5).

**Assessing the immune modulatory impact on collagen architecture.** For the high- and low-risk patients identified by CollaTIL in the D0-D2 cohorts, we compared the TIL density and entropy value of the disorder in collagen fiber orientations. TIL density is defined as the number of TILs divided by the total number of nuclei in the stromal regions of the invasive tumor front compartments on the H&E WSI. The disorder of collagen fiber orientations was measured using the mean entropy value of collagen fiber orientation disorder within a 200x200-pixel tumor neighborhood.

**Identification of high confidence somatic mutations in the TCGA dataset.** Somatic single-nucleotide mutation calls (SNVs and indels), identified in whole-exome sequencing (WES) studies of the HGSOC, CSCC, and EC tumor types, were obtained from GDC platform (MC3 V.0.2.8, https://synapse.org/MC3)[21]. The MC3 effort provided consensus calls of variants from multiple genomic platforms including, MuTect, VarScan2, Somatic Sniper, MuSE, Radia, Indelocator, and Pindel. We extracted the position and nucleotide change information for all single-nucleotide somatic mutations, which are integrated into a combined MAF file. We then categorized aberrations as synonymous (*Wt*) and non-synonymous (*Mut*). Missense, Frame shift deletion, Nonsense, Inframe deletion, Inframe insertion, Splice site, Frame shift insertion, Translation start site, Nonstop mutations were marked as non-synonymous (*Mut*) mutations, while Silent, 3′UTR, RNA, Intron, 3′Flank, 5′UTR, 5′Flank were labeled are synonymous (*Wt*) group. We additionally annotated a single unique mutation entry having multiple annotations belonging to different variant classification as non-synonymous, where at least one of the multiple entries of the annotations shows in the non-synonymous category. We performed extensive filtering to minimize sequencing artifacts, mutation calling errors and to exclude likely false-positive mutations. In fact, mutation calls with tumor depth or normal depth less than 30 reads, and tumor frequency less than 0.1 or normal frequency greater than 0.05 were filtered out from our analysis. We also excluded hypermutated samples, as they likely reflect distinct underlying mutational processes, and they tend to have an adverse effect on statistical power. We identified hypermutated samples as having more mutations than 1.5 times the interquartile range above the third quartile of samples within the same cancer cohort.

**Differential gene expression analysis pipeline.** To identify differentially expressed genes between two groups of high-risk and low-risk patients a series of bioinformatics analyses were performed. The analysis pipeline began with raw read counts downloaded from GDC portal, which were quality checked and trimmed. The trimmed reads were aligned to the GRCh38 human reference genome[22]. The resulting count data were then normalized and compared between the two groups using DESeq2 to identify differentially expressed genes. This analysis was performed on all patients who had a RNASeq data and H&E WSI. The results were compared to generate a comprehensive list of differentially expressed genes across all samples. This approach enabled the identification of genes that were differentially expressed between the two groups of high-risk and low-risk patients.

**Ethical statement.** This study was performed under the Emory University Institutional Review Board (IRB) protocol STUDY00005888, which was approved as a non-human study and all relevant ethical regulations were followed. De-identified human samples were obtained from UH and CCF, collected under the same IRB approved protocol STUDY00005888. UH and CCF collected specimens with participants' informed consent.

**Reporting summary.** Further information on research design is available in the Nature Portfolio Reporting Summary linked to this article.

## Results

**Prognostic role of CollaTIL.** The Cox regression model trained on the D0 cohort consisted of 14 CollaTIL features that were found to be associated with OS, containing features from both immune and collagen components within the TME (Table 1). CollaTIL was associated with OS in validation cohorts (D1-D4) and with PFS in validation cohorts (D5-D8). Specifically, in the chemotherapy-treated validation cohorts (D1, D3 and D8), it was associated with OS in D1 ($p = 0.0256$, HR = 2.54, 95% CI = 1.21–5.34, c-index = 0.72), D3 ($p = 0.0213$, HR = 7.36, 95% CI = 2.73–19.8, c-index = 0.7), and with PFS in D8 ($p = 0.0434$, HR = 3.07, 95% CI = 1.01–9.85, c-index = 0.74). In the radiotherapy-treated validation cohort (D2), it was also associated with OS ($p = 0.006$, HR = 2.87, 95% CI = 1.49–5.53, c-index = 0.72). Furthermore, the signature identified high risk patients who had significantly worse PFS than low-risk patients in the immunotherapy-treated validation cohorts (D5, D6, and D7, $p = 0.0184$, HR = 2.72, 95% CI = 1.36–5.42, c-index = 0.65) (Fig. 3). The results also indicate that the CollaTIL signature outperformed other signatures derived from either Collagen or Immune features alone, highlighting the importance of their combination (Supplementary Methods).

**Assessing the immune modulatory impact on collagen architecture.** To assess the immune modulatory impact on collagen architecture, we examined the D0-D2 cohorts. We conducted a comparison of the entropy value of collagen fiber orientation disorder and TIL density between the high- and low-risk patients identified by CollaTIL. The findings indicate that low-risk patients exhibit high TIL density and higher disorder in collagen fiber orientations compared to high-risk patients (Fig. 4).

**Association of CollaTIL with prognostic gene signatures, mutated genes, and metabolic pathways.** To elucidate the association and prognostic implications surrounding CollaTIL and the presence of mutated genes in HGSOC and CSCC patients (D0-D2), we systematically categorized a subset of patients of the D0-D2 cohorts. Specifically, we considered patients with available mutation annotations and H&E WSI. These patients were divided into two distinct groups, namely those harboring gene mutations and those exhibiting wild-type gene status. Through our analysis, we have revealed that nonsynonymous somatic mutations in PIK3CA were not associated with risk in CSCC patients (D1 and D2). However, they were linked to a diminished risk of recurrence or death in european american (EA) patients who were treated with chemotherapy (Fig. 5a).

**Table 1 Top features contributing to CollaTIL signature.**

| Feature index | Feature description | HR (per unit increase) |
|---|---|---|
| 1 | Ratio of non-TILs density to the surrounding (20 microns proximity) TILs in the epithelium compartment | 1.17 |
| 2 | Number of epithelial TIL clusters surrounding (20 microns proximity) a non-TIL cluster in the epithelium compartment | 0.95 |
| 3 | Presence percentage (ratio of present clusters to total number of clusters) of stromal non-TIL clusters being around another non-TIL cluster in the stromal compartment | 1.26 |
| 4 | Intersected area of clusters of epithelial TILs and non-TILs in invasive tumor front compartment | 0.51 |
| 5 | Minimum area of stromal TIL clusters in invasive tumor front compartment | 1.78 |
| 6 | Range of area of epithelial non-TIL clusters in invasive tumor front compartment | 2.26 |
| 7 | Mean entropy value of the collagen fiber orientation disorder feature map using 200x200-pixel neighborhood | 0.75 |
| 8 | Minimum entropy value of the collagen fiber orientation disorder feature map using 200x200-pixel neighborhood | 0.5 |
| 9 | Maximum entropy value of the collagen fiber orientation disorder feature map using 250x250-pixel neighborhood | 0.94 |
| 10 | Minimum entropy value of the collagen fiber orientation disorder feature map using 350x350-pixel neighborhood | 0.84 |
| 11 | Minimum entropy value of the collagen fiber orientation disorder feature map using 400x400-pixel neighborhood | 0.64 |
| 12 | Minimum entropy value of the collagen fiber orientation disorder feature map using 450x450-pixel neighborhood | 1.64 |
| 13 | Maximum entropy value of the collagen fiber orientation disorder feature map using 550x550-pixel neighborhood | 1.38 |
| 14 | Maximum entropy value of the collagen fiber orientation disorder feature map using 600x600-pixel neighborhood | 0.36 |

We also aimed to pinpoint genes that could provide insight into the underlying biological mechanisms associated with CollaTIL in CSCC patients (D1-D2). To achieve this, we initially identified genes whose expressions exhibited significant correlations with estimated risk scores, as predicted by the pathomic-based model of TME (Pearson correlation $P < 0.05$). Subsequently, we performed an analysis of differentially expressed genes to identify the most significant candidates among patients predicted to be at high and low risk according to the model, selected using a false discovery rate (FDR)[23] threshold of <0.05 and a log2-fold change threshold of >1. Intriguingly, our investigation unveiled two distinct sets of genes which exhibited significant correlations with risk and differential expression patterns within the high- and low-risk groups of CSCC patients in D1 and D2 cohorts, respectively.

We then employed single-sample gene set enrichment (ssGSEA) methodology to estimate the activity index (Supplementary Methods) of the identified gene signatures[24]. The results revealed that a gene signature including *COX6A1, COX7B, ATP5H, COQ10A, NDUFA1, NDUFA5, NDUFA6, NDUFA8, NDUFA12, and NDUFC1* belonging to the TCA-cycle was significantly down-regulated in high-risk CSCC patients, as determined by Wilcoxon *p*-value comparison between high- and low-risk groups predicted by the model ($P = 0.0026$) (Supplementary Methods, Fig. 5b).

In a similar way, among the genes identified in HGSOC patients (D0) (Supplementary Data 1), gene sets of *CARNMT1, DUOX1, HIBADH, OAT, PSMA7, PSMC2, PSMC4, PSMD11, PSMD8, RPL28, RPS16, RPS19, SERINC4, SLC6A11, AASS, and AIMP2*, belonging to amino-acid, and gene sets of *ADORA3, ATP8B4, C1QB, C3AR1, C5AR1, CD163, CD300A, FCGR2A, LIPA, LY96, MSR1, SLCO2B* signatures, were identified. Using ssGSEA, we compared these gene signatures and found that the up-regulation of genes belonging to amino-acid and macrophage gene signatures are significantly associated with a higher risk score predicted by the pathomic-based model of TME (PAmino-acid = 0.0023, PMacrophage = 0.001) (Supplementary Data 1 and Fig. 5c, d).

Furthermore, we conducted an in-depth exploration of the association between the top identified genes in CSCC and HGSOC and patient overall survival. Remarkably, among the top genes, in addition to some important well-known genes for each of the cancer types, we also identified three genes *APOBEC3H, KIRREL1, and FAM166B*, that are significantly correlated with OS for CSCC and HGSOC patients, respectively. These intriguing findings suggest that the expression of these genes may serve as a prognostic factor for cancer patient survival (Fig. 5e–g).

**Protein expression modulation and genetic potential analysis associated with CollaTIL.** In HGSOC patients (D0), we found that while gene expression of identified gene sets belonging to macrophages (*VSIG4, ATP9A, CD163, C3AR1, C1QB*) and amino-acids (*PSMC2, PSMC4, PSMA7, PSMD8, PSMD11*) did not show significant differences at the transcript levels compared to the normal tissue, however, the protein expression of *VSIG4, ATP9A, CD163, C3AR1, and C1QB*, was significantly down-regulated in primary tumors as compared to normal tissue (Fig. 6a–e). Moreover, the protein expression of a subset of the genes belonging to amino acid (*PSMC2, PSMC4, PSMA7, PSMD8, and PSMD11*) were significantly upregulated in primary tumors as compared to normal tissue (Fig. 6f–i).

To further investigate the functional potentials of genes or proteins exhibiting significant changes in expression compared to other low-risk group in HGSOC, we utilized the databases of Broad's Achilles and Sanger's SCORE projects[25,26]. These projects assess gene effects through CRISPR knockout experiments conducted across diverse HGSOC cell lines[27]. Our findings showed that knocking out extremely significant genes belonging to macrophages (*C3AR1 and C1QB*) had a low negative gene score, indicating less impact on cellular growth (Supplementary Fig. 1a, b). On the other hand, knocking out Amino acid associated genes such as *PSMC4 and PSMA7* showed more negative gene value, suggesting strong impact on cell growth or cell death (Supplementary Fig. 1c, d). These concurrent findings suggest the presence of a prognostic and therapeutic significance

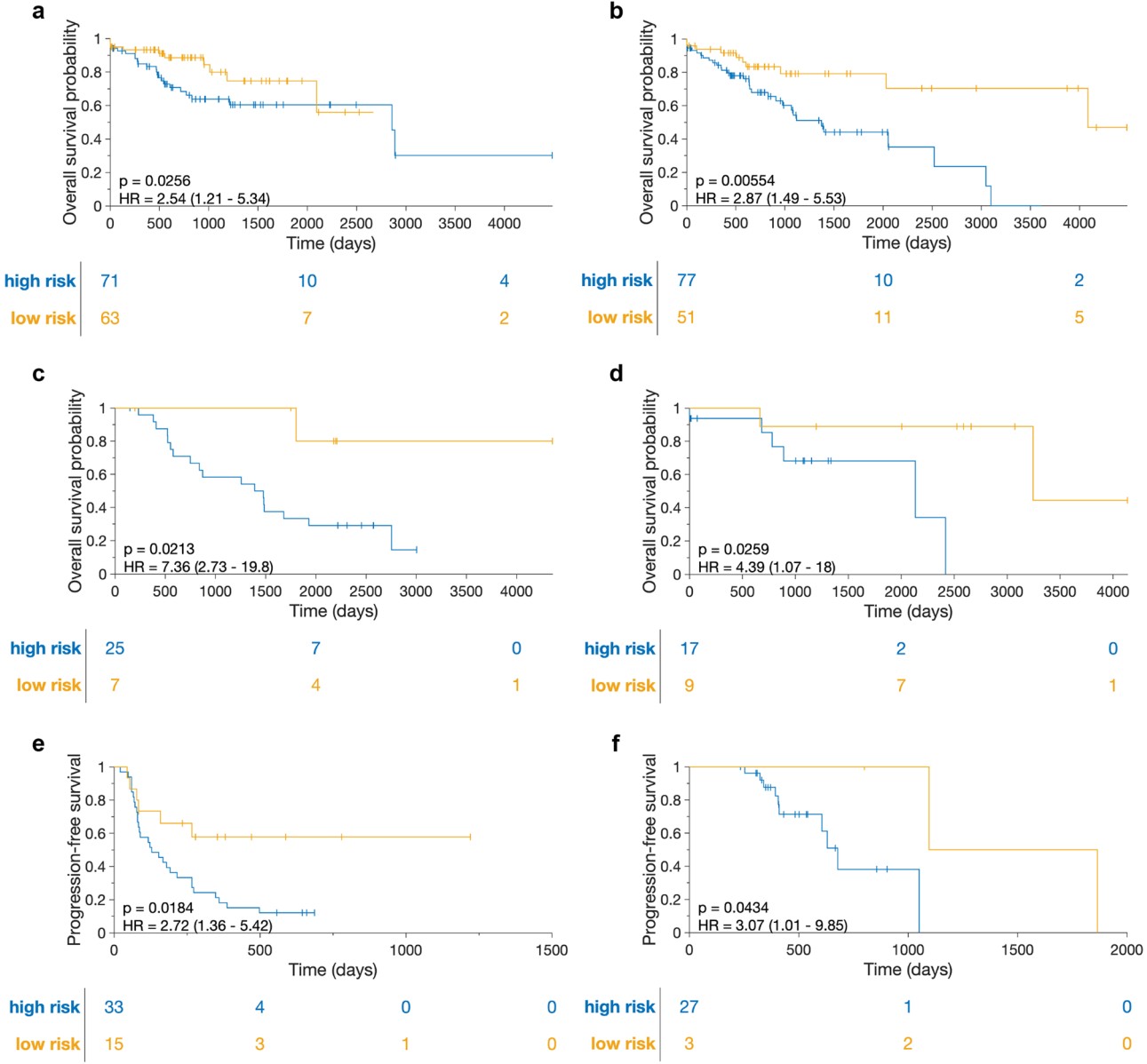

**Fig. 3 Kaplan–Meier curves on validation cohorts (D1-D8) using CollaTIL model.** Shown are Kaplan–Meier estimates of OS and PFS for high-risk patients as compared to their low-risk counterparts in **a** cohort of CSCC patients treated with chemotherapy (D1, $n = 134$). **b** cohort of CSCC patients treated with radiotherapy (D2, $n = 128$). **c** cohort of EC patients treated with chemotherapy (D3, $n = 32$). **d** cohort of EC patients not treated with chemotherapy (D4, $n = 26$). **e** cohorts of HGSOC, CSCC, and EC patients treated with immunotherapy in the recurrent setting (D5, D6, D7, $n = 48$). **f** cohort of HGSOC patients treated with chemotherapy (D8, $n = 30$). The statistical significance of differences in survival rates between high-risk and low-risk groups was determined using the log-rank test (P). EC endometrial carcinoma, HGSOC high grade serous ovarian carcinoma, CSCC cervical squamous cell carcinoma, OS overall survival, PFS progression-free survival.

of gene sets belonging to the macrophages and amino acids in HGSOC.

## Discussion

Immune cells occupy a pivotal position in the body's defense against cancer, with TILs emerging as key players in the anti-tumor immune response[5,6]. Notably, TILs secrete cytokines that stimulate immune activity and recruit additional immune cells to the tumor site, strengthening the anti-tumor response[5]. Moreover, TILs possess the unique ability to recognize and target cancer cells by identifying specific antigens displayed on the surface of malignant cells[28]. The presence of TILs within tumors is consistently linked to improved prognosis and enhanced responses to immunotherapy, highlighting their

crucial role in combating cancer[29]. Consequently, unraveling the intricate mechanisms of TIL function and developing strategies to potentiate their anti-tumor activity are of utmost importance, holding potential to revolutionize cancer treatment paradigms[29,30].

A pivotal aspect of TIL function lies in their ability to modulate the ECM, a complex network of proteins and carbohydrates that provide structural integrity to tissues[7,9,10]. TILs exert their influence on the ECM by secreting matrix metalloproteinases, a family of enzymes capable of breaking down various ECM components, including collagen, gelatin, and other matrix proteins[8]. This leads to the question of whether the heightened immune cell response correlates with elevated enzymatic activity, leading to more degradation of collagen architecture and the

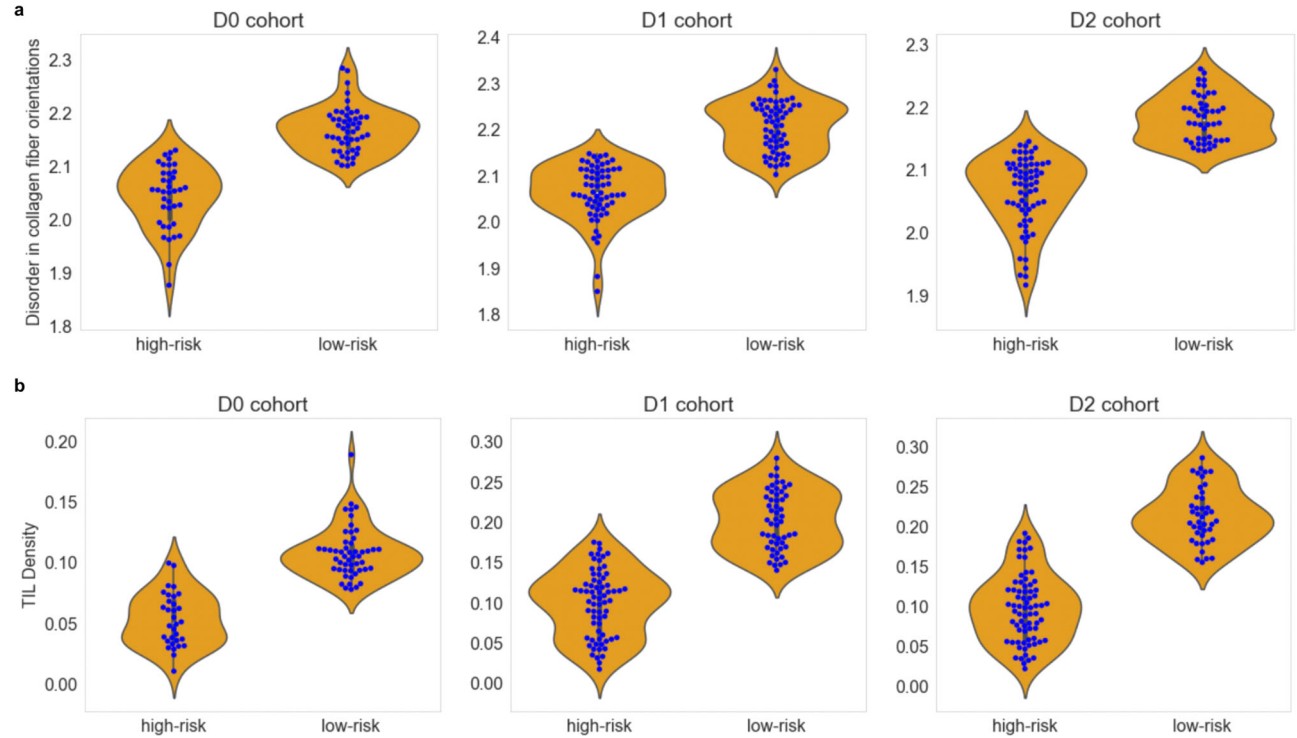

**Fig. 4 Assessing the immune modulatory impact on collagen architecture in HGSOC (D0) and CSCC (D1-D2) patients. a** Shown are the mean entropy values of the disorder in collagen fiber orientations in the high- and low-risk groups identified by CollaTIL in HGSOC (D0, $n = 95$) and CSCC (D1 and D2, $n = 262$). **b** Shown are the TIL density values in the high- and low-risk groups identified by CollaTIL in HGSOC (D0, $n = 95$) and CSCC (D1 and D2, $n = 262$). HGSOC high grade serous ovarian carcinoma, CSCC cervical squamous cell carcinoma, TIL Tumor-infiltrating lymphocytes.

subsequent formation of more disordered collagen fibers. Moreover, this prompts further speculation on whether immune cells assume an alternative anti-tumor protective role by disrupting collagen fibers, a potential conduit for tumor progression and metastasis[3,4]. The primary objective of this study was to employ advanced computational analysis and machine learning tools to quantitatively (1) ascertain whether on H&E images, the presence of an increased immune infiltrate is associated with a more disrupted and chaotic collagen architecture and conversely whether a subdued immune response is associated with a more coherent collagen organization and (2) whether features relating to the immune-collagen spatial relationship are associated with clinically relevant outcomes.

In this study, we quantitatively characterized the immune and collagen architecture on 493 H&E WSIs obtained from surgically-resected neoplasms to predict cancer prognosis in three types of gynecologic cancers: HGSOC, CSCC, and EC, treated with various adjuvant and recurrence therapies. CollaTIL quantitatively captures the architecture of two important components within the TME: immune milieu and collagen fibers. Unlike previous approaches that focused on analyzing a single component within the TME for cancer prognosis[11,12,31], CollaTIL offers a more comprehensive understanding of the TME by integrating information from these two components. CollaTIL features comprise of the characterization of the architecture of TILs and their interplay with surrounding cells, as well as the assessment of the degree of disorder in collagen fiber orientations within the stromal regions of H&E WSI. The CollaTIL features showed associations with OS in the validation cohorts (D1-D4) and with PFS in the validation cohorts (D5-D8). We observed that low-risk patients identified by CollaTIL exhibit high TIL density and higher disorder in collagen fiber orientations, while high-risk patients identified by CollaTIL show low TIL density and lower disorder in collagen fiber orientations.

Beyond assessing the morphologic basis of the immune modulatory impact on collagen architecture, we also delved into the molecular basis through genomic analysis. We identified gene signatures that could provide insight into the biological mechanisms associated with CollaTIL in HGSOC (D0). Among the top genes identified in HGSOC patients (D0), a subset of Amino-acid and Macrophage gene signatures were found to be significantly associated with a higher risk score predicted by CollaTIL. Amino acid metabolism is essential for cancer cell growth and survival. Dysregulation of amino acid metabolism can lead to changes in the expression of genes involved in cancer development and progression[32]. On the other hand, pro-tumoral macrophages promote tumor growth and progression by suppressing the immune system and promoting angiogenesis, and the infiltration of pro-tumoral macrophages into HGSOC tumors is associated with poor prognosis. Therefore, they are known to play important roles in predicting cancer prognosis of HGSOC patients[33,34].

It's important to note that amino acid metabolism is not only vital for cancer cells but also for macrophage generation and their polarization into M1 or M2 phenotypes[35]. M1 macrophages contribute to ECM degradation and bolster an anti-tumor immune response, whereas M2 macrophages promote ECM remodeling that facilitates tumor progression[36]. These results indicate the possibility of enhanced presence of pro-tumoral macrophages (M2) in high-risk HGSOC patients, where patients with identified amino acid signatures may show overall weakened immune activity and a positive correlation with angiogenesis-associated genes.

Furthermore, we identified gene signatures associated with CollaTIL in CSCC (D1-D2). Our findings revealed that in CSCC patients, the downregulation of TCA-cycle genes such as *COX6A1*, *COX7B*, *ATP5H*, *COQ10A*, *NDUFA1*, *NDUFA5*, *NDUFA6*, *NDUFA8*, *NDUFA12*, and *NDUFC1* may be related to

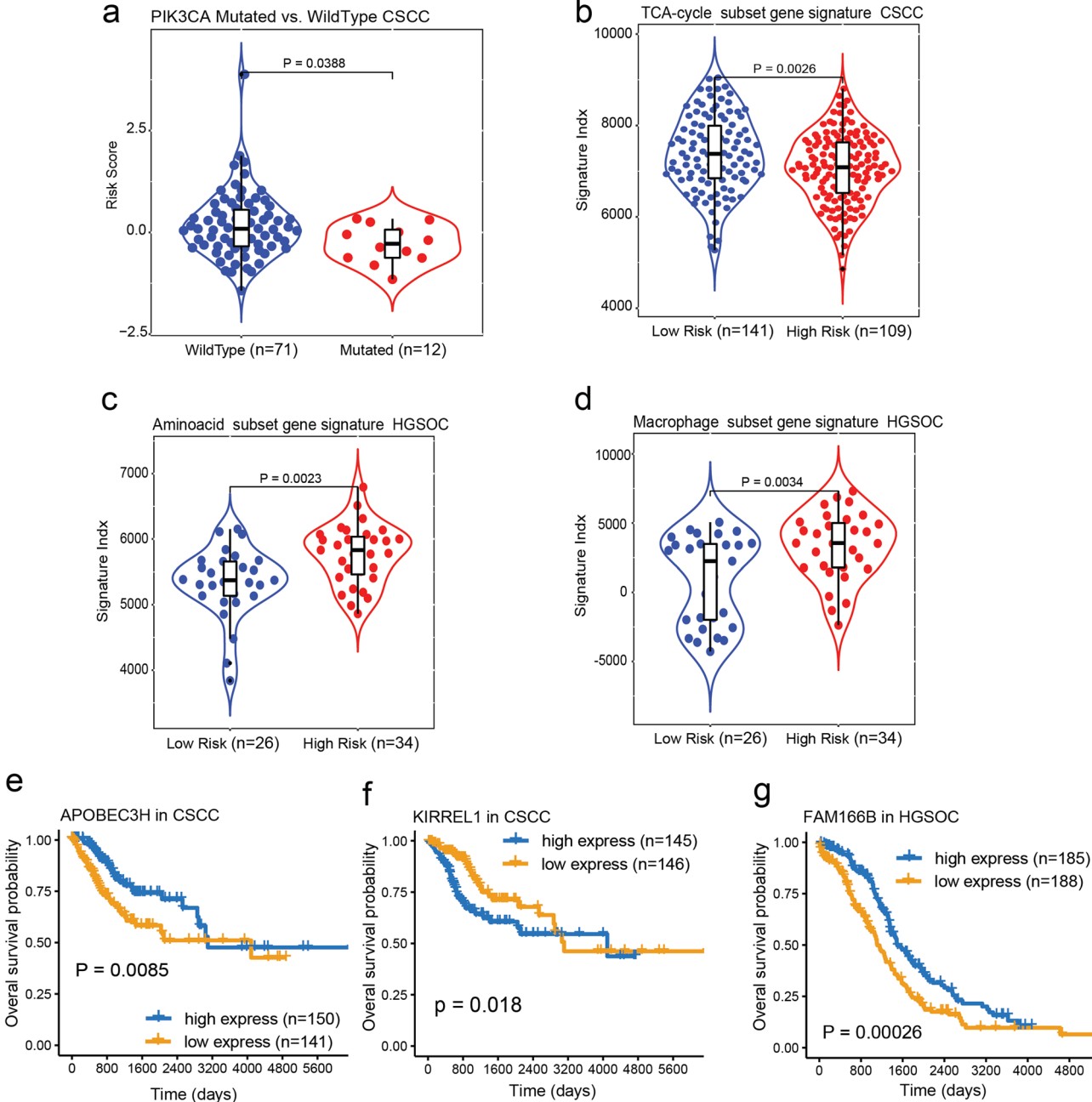

**Fig. 5 Genomics association and prognostic significance of CollaTIL in CSCC and HGSOC. a** Box-violon plot depicting significant difference of the predicted risk between CSCC patients harboring mutation in PIK3CA as compared to wildtype counterparts. **b–d** Box-violons plots detailing the signature activities of TCA-Cycle in CSCC, and Amino acid and Macrophage in HGSOC for high-risk patients compared to the low-risk patients. Statistical significance of differences in the signature activities was estimated using a Wilcoxen signed rank test. **e–g** OS estimates of CSCC (**e**), CSCC (**f**) and HGSOC (**g**) patients. Shown are Kaplan–Meier estimates of OS for patients with high expression of APOBEC3H in CSCC, KIRREL1 in CSCC and FAM166B in HGSOC as compared to their low expressed counterparts. The statistical significance of differences in survival rates between high expressed and low expressed categories was determined using the LogRank test (P). The Violin plots represent the probability density of patients at different risk scores or signature indices. Embedded Boxplots summarize the distribution, indicating the median (center line), quartiles (box edges), and potential outliers (beyond whiskers). HGSOC high grade serous ovarian carcinoma, CSCC cervical squamous cell carcinoma, OS overall survival, PFS progression-free survival.

molecular changes in cancer cells, reflecting a shift in metabolic pathways[37]. These genes are part of the TCA-cycle gene signature, which has been previously linked to predicting cancer prognosis in CSCC patients[38]. The downregulation of these TCA-cycle genes can affect cellular metabolism, energy production, and redox balance, which in turn could influence ECM remodeling processes. This raises the question for future interrogation on whether the downregulation of TCA-cycle genes influence the

remodeling of the ECM. Future work will focus on understanding the potential relationship between TCA-cycle gene expression and ECM remodeling processes.

We also identified three genes, *APOBEC3H*, *KIRREL1*, and *FAM166B*, significantly correlated with OS for CSCC and HGSOC patients, respectively, which suggest that the expression of these genes could be a prognostic factor for patient survival. Additionally, we identified several highly relevant genes in CSCC

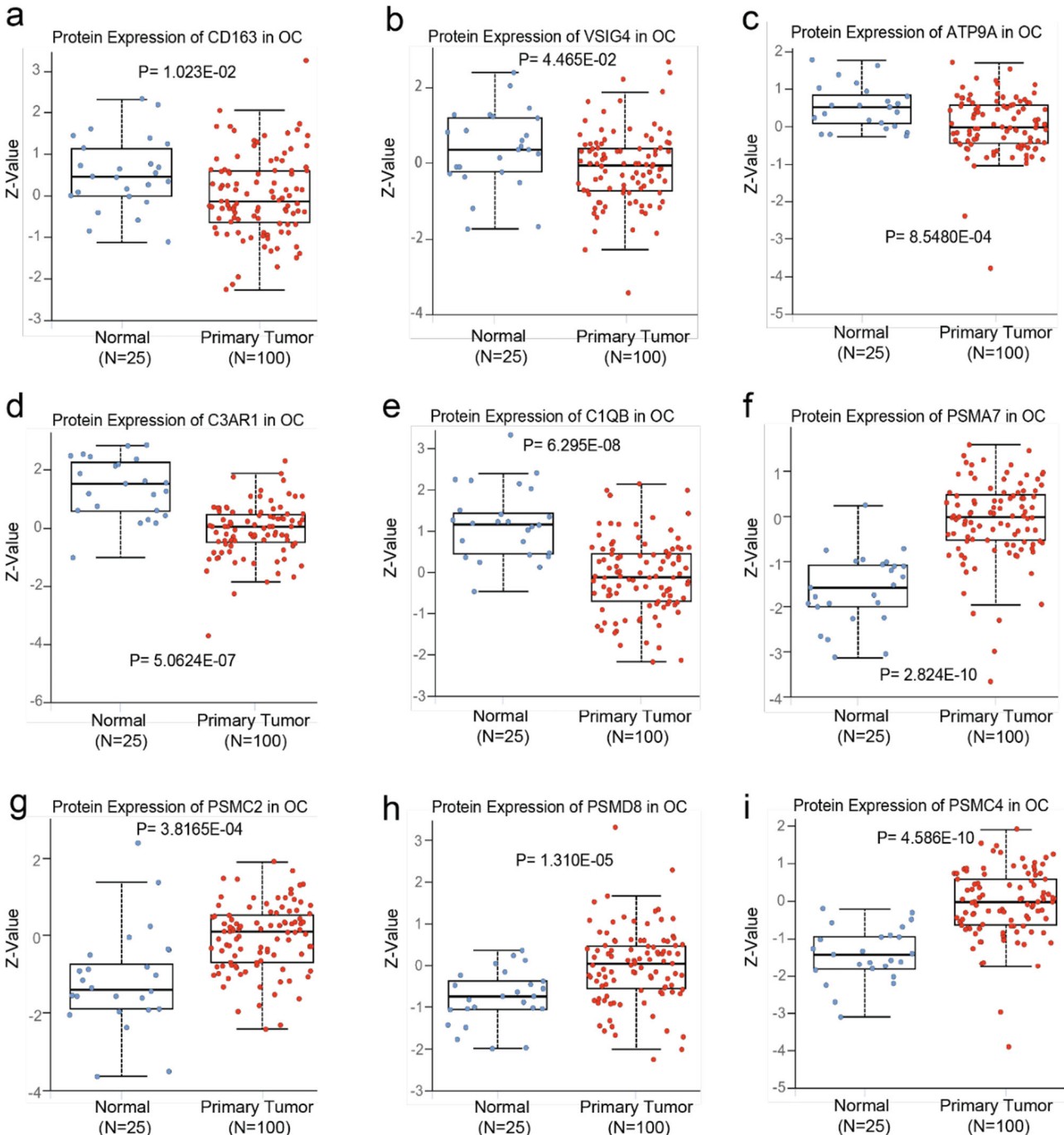

**Fig. 6 Protein expression comparison between tumor and normal ovary tissues of the identified gene sets belonging to macrophage and Amino-acid on CPTAC dataset. a–e** Protein expression of Macrophages associated gene sets *VSIG4, ATP9A, CD163, C3AR1, and C1QB*, were significantly downregulated in primary tumors compared to the normal tissue in HGSOC. **f–i** Protein expression of amino acid associated gene sets PSMA7, PSMC2, PSMC8, and PSMD4, were significantly upregulated in primary tumors compared to the normal tissue in HGSOC. Log2 Spectral count ratio values from CPTAC were first normalized within each sample profile, then normalized across samples. *Z*-values represent standard deviations from the median across samples for the given cancer type. *P* indicates the *p*-value obtained from the statistical *t*-test used to compare the two groups of patients. The Boxplots summarize the distribution, indicating the median (center line), quartiles (box edges), and potential outliers (beyond whiskers). HGSOC high grade serous ovarian carcinoma, CPTAC clinical proteomic tumor analysis consortium.

patients (D1 and D2) whose expression is strongly correlated with the risk scores predicted based on CollaTIL (Supplementary Fig. 2). This finding supports the notion that the expression of some of these genes can have a significant impact on the progression of the disease and can be used as predictive markers for prognosis. Interestingly, our analysis of TCA-related genes in CSCC patients revealed that four TCA-related genes, *NDUFA1,*

*NDUFA12, COX7B,* and *CD8A,* were significantly upregulated and associated with a better prognosis specifically in CSCC patients (Supplementary Fig. 3).

We acknowledge that our study had certain limitations. One notable limitation was the absence of specialized staining techniques, such as immunohistochemistry (IHC) or Verhoeff-Van Gieson (EVG) staining, which would have enabled a quantitative

analysis of collagen fiber segmentation. This limitation stemmed from the unavailability of IHC and EVG-stained whole slide images for the cohorts used in our study. Moreover, the interplay between immune and collagen components within the TME can be bi-directional. Previous studies have suggested that collagen density can influence the activity of TILs in the TME, influencing their ability to kill cancer cells[39,40]. Additionally, we did not consider different types of immune cells in the TME, such as tumor-associated macrophages, which also play a critical role in ECM remodeling[41]. Future work will focus on exploring the roles of different types of immune cells in the TME and their impact on collagen architecture.

In conclusion, our study identified prognostic features from the immune and collagen components within the TME in patients with CSCC, EC, and HGSOC. These features were associated with survival outcomes in different gynecologic cancer cohorts treated with various adjuvant and recurrence therapies. The CollaTIL framework enables the extraction of features that are able to capture the complex relationship between immune and collagen architecture within the TME. These features, in turn, enable accurate risk stratification of gynecologic cancers. Future work will involve assessing whether these features could also aid in the selection of appropriate treatment strategies for gynecologic cancer patients.

## Data availability

All data required to reproduce the results presented here are available at https://github.com/arp95/collatil_biomarker_gyn[42]. D0, D1, and D2 cohorts were generated by TCGA Research Network (http://cancergenome.nih.gov/) and they have made them publicly available[13]. D8 cohort was generated by Memorial Sloan Kettering Cancer Center and is made publicly available at Synapse (Sage Bionetworks) under accession code syn25946117[14]. Since the cases from the involved institutions are protected through institutional compliance, the clinical repository of cases can only be shared per specific institutional review board (IRB) requirements. Upon reasonable request, a data sharing agreement can be initiated between the interested parties and the clinical institution following institution-specific guidelines. This applies to the cohorts known as D3, D4, D5, D6, and D7. For inquiries or requests regarding data sharing, please contact the corresponding author. Source data underlying the graphs in Figs. 3–6 is available at https://github.com/arp95/collatil_biomarker_gyn.

## Code availability

All code required to reproduce the results presented here is available at https://github.com/arp95/collatil_biomarker_gyn, along with relevant documentation[42].

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

## Acknowledgements
Research reported in this publication was supported by the National Cancer Institute under award numbers R01CA249992-01A1, R01CA202752-01A1, R01CA208236-01A1, R01CA216579-01A1, R01CA220581-01A1, R01CA257612-01A1, 1U01CA239055-01, 1U01CA248226-01, 1U54CA254566-01, National Heart, Lung and Blood Institute 1R01HL15127701A1, R01HL15807101A1, National Institute of Biomedical Imaging and Bioengineering 1R43EB028736-01, National Center for Research Resources under award number 1 C06 RR12463-01, VA Merit Review Award IBX004121A from the United States Department of Veterans Affairs Biomedical Laboratory Research and Development Service the Office of the Assistant Secretary of Defense for Health Affairs, through the Breast Cancer Research Program (W81XWH-19-1-0668), the Prostate Cancer Research Program (W81XWH-15-1-0558, W81XWH-20-1-0851), the Lung Cancer Research Program (W81XWH-18-1-0440, W81XWH-20-1-0595), the Peer Reviewed Cancer Research Program (W81XWH-18-1-0404, W81XWH-21-1-0345, W81XWH-21-1-0160), the Mayo Clinic Breast Cancer SPORE grant P50 CA116201 from the NIH, the Kidney Precision Medicine Project (KPMP) Glue Grant, and sponsored research agreements from Bristol Myers-Squibb, Boehringer-Ingelheim, Eli-Lilly and AstraZeneca. The content is solely the responsibility of the authors and does not necessarily represent the official views of the National Institutes of Health, the U.S. Department of Veterans Affairs, the Department of Defense, or the United States Government.

## Author contributions
A.A., S.K., and A.M. conceptualized and designed the study. A.A., S.K., D.B., H.L., S.E., and G.C. were responsible for the experiment run. S.A. and H.M. were responsible for providing the image data and corresponding clinical information. M.M. and S.A. were responsible for quality check of intermediate masks. S.K. and D.B. were responsible for the analysis of underlying biological pathways. P.F. supervised the statistical analysis. E.T., S.M., T.P., and S.A. were responsible for providing the biological interpretation of the findings. The manuscript was written primarily by A.A., S.K., and A.M., with support from all authors. All authors have read and approved the manuscript.

## Competing interests
The authors declare the following competing interests: A.M. is an equity holder in Picture Health, Elucid Bioimaging, and Inspirata Inc. Currently he serves on the advisory board of Picture Health, Aiforia Inc, and SimBioSys. He also currently consults for SimBioSys. He also has sponsored research agreements with AstraZeneca, Boehringer-Ingelheim, Eli-Lilly and Bristol Myers-Squibb. His technology has been licensed to Picture Health and Elucid Bioimaging. He is also involved in 3 different R01 grants with Inspirata Inc. He also serves as a member for the Frederick National Laboratory Advisory Committee.
