## [Peer Review File · Communications Medicine]

Reviewers' comments:

Reviewer #1 (Remarks to the Author):

The study of Aggarwal et al. is investigating the role of collagen fibres and TILs in several gynecological malignancies. The study bears novelty as it integrates ECM elements into the computational perspective on malignancies. The study uses publicly available data sets from TCGA and hospital derived data sets. The cases were characterized regarding genetics and protein expression.

The study might be improved in these minor points

- There are many studies out for TIL evaluation with conventional digital pathology, but also computational models. It would be interesting to see more data about the "added value" achieved with the integration of the collagen part. This could be plotted e.g. with ROC curve comparisons
- Was there an image-based groundtruth achieved for the collagen part, e.g. is the algorithm able to distinguish elastic fibers from collagen fibres. Did an overlay with histochemical (EVG) or immunohistochemical images produce correct results.
- The tile format of 3000x3000 seems quite big in relation to the case numbers. Shouldn't the performance not be tested on larger validation sets? Please indicate the total number of tiles accessible for training and validation.
- Clinically relevant molecular signatures/characteristics would be of interest. In endometrial carcinoma the molecular subtypes would be interesting, in cervix carcinoma the HPV dependency, in ovarian carcinoma the HRD status.

Reviewer #2 (Remarks to the Author):

The authors suggest a hypothesis that a combination of TIL TME composition and collagen fiber structure plays a role in determining patient outcome in several gynecological cancers. Across several cohorts they show how disorganized fibers correspond to higher degree of immune infiltration, while regular fiber structure to lower immune infiltration and worse patient outcomes. TME analysis is followed by genome, gene expression and protein level analysis in a subset of cohorts.

I find the idea of analyzing TME fibers together with the TIL composition very interesting and worth investigating. While their correspondence is well presented and their effect in patient groups is demonstrated, I find several drawbacks of the analysis and result presentation in the manuscript. Additionally, the molecular data analysis lacks any link to the WSI analysis and is presented in a way that is hard to follow or comprehend with relationship to the main finding of the paper (TIL-fiber relationship).

Major points

1. Across the paper, the cohorts, diseases, datasets are not well described. The results section

should start by describing D0-D8, with their specific characteristics and content. Otherwise, the selection of cohorts for different parts of the analysis seems arbitrary and is hard to understand. Why, for example, D0 is the training cohort?

2. I see no link between genome/gene/protein level analysis with the collagen/TIL analysis. These seem like two separate stories. What is the novelty in the findings in the molecular data? What is the link between the “Amino acid and Macrophage gene signatures” and TILs/collagen fiber composition?

3. Is CollaTIL a tool? I didn't find any link to a code repository. In that case the manuscript presents an analysis, it shouldn't have a name.

4. The molecular data analysis is unordered and hard to follow. The results should be selected in a way to build a coherent story that links the following parts together.

Detailed points

L78-80 “previous findings..” lack citation

L125-126 An existing pretrained deep learning model was used to segment the epithelial and stromal regions on these tiles. The authors should explicitly mention it here that these results were additionally validated by pathologists. I only found this information while scrolling through the Supplementary Material.

L133 “derivative-of-Gaussian based model” is not self-explanatory. Can you explain what is it and how is it used to “capture the fiber orientations by detecting linear structures within these regions”.

L183 What kind of molecular data? Why wouldn't you make this comparison in cohorts D1-D8?

Fig 5. Why wouldn't you combine these plots? Cohorts can be marked with different colors in a single plot.

Fig 6. What is “Signature index”?

L204 What are “risk scores” how are they estimated and what are “race-specific architectural-based models of TME”? There is no citation here either.

L207 Abbreviation FDR is not explained.

L209 What are the “two distinct set of genes”? Are these different sets of genes for cohort D1 and D2 (in that case add “respectively” at the end of the sentence). In any case those “sets of genes” need to be explained.

L213 “activity index” is not self-explanatory, what is its meaning?

L223 SSGSEA, or ssGSEA (few lines before)?

L227 What do you mean by “top identified genes”? Based on what ranking are these top genes?

L237-245 What do these genes mean, are they related to TILs or fibers?

L246 “highly altered genes” – do you mean here expression, protein level or mutations?

L246-255 What is the reason to verify impact on cellular growth here? I don’t see what is the logical thread here.

Fig. 8. What is “Chronos”?

L297 “low-risk patients identified by CollaTIL” – does your method identify the patients or rather features related to the OS and PFS?

L306 “Amino acid and Macrophage gene signatures..” how are these genes related to the results of WSI analysis? Or is it 2 separate analyses?

L381 Why was the model trained on “oral cavity tissue microarray images”? What was the test set performance?

L385 How was the threshold “fine-tuned”?

L388 How were the regions selected? Manually?

L406 Why not use Hovernet cell classification results to identify TILs? How accurate is the classification method proposed by the authors?

L423 Why bins are from 0 to 17? What do they correspond to and why 17?

L451 I don’t see which 7 features these are in the Supplementary Table 1. There are 6 TIL-related features in the table.

L464 “CollaTIL-defined low- and high-risk patients” – again are these patients defined by CollaTIL? How?

L506 In my opinion, with the publication the datasets should be public. Otherwise, we cannot warranty reproducibility of this research.

Wallace H. Coulter Department of
Biomedical Engineering

Paper number: COMMSMED-23-0509-T

Communications Medicine

Title: Computational Pathology Unveils Immune-Mediated Collagen Disruption: Predicting Clinically Relevant Outcomes in Gynecologic Malignancies

Dear Reviewers,

Please find enclosed a revised version of our manuscript “**Computational Pathology Unveils Immune-Mediated Collagen Disruption: Predicting Clinically Relevant Outcomes in Gynecologic Malignancies**”. The modifications introduced in this new version have been color-coded to facilitate comparison with the original submission. Specific responses are in dark red italics while modifications to the manuscript are in blue.

We believe we have addressed the comments and have updated our manuscript accordingly. Detailed responses are outlined below. We hope that this new version satisfactorily addresses the concerns. Thank you for your consideration.

Best regards,

Anant Madabhushi, PhD, FAIMBE, FIEEE, FNAI

Robert W. Woodruff Professor

Wallace H. Coulter Department of Biomedical Engineering,
Radiology and Imaging Sciences, Biomedical Informatics, Pathology, Urology.

Georgia Institute of Technology and Emory University

Research Career Scientist

Atlanta Veterans Administration Medical Center

Health Sciences Research Building II

1750 Haygood Drive, Suite N647

Atlanta, Georgia 30322

E-mail: anantm@emory.edu

Youtube Video: <https://tinyurl.com/y987ghao>

Wiki: https://en.wikipedia.org/wiki/Anant_Madabhushi

Twitter: [@anantm](https://twitter.com/anantm)

Summary of Major Changes

- **CollaTIL Tool:** We have provided a link to the CollaTIL code repository on GitHub (https://github.com/arp95/collatil_biomarker_gyn), confirming that CollaTIL is indeed a tool used in the study.
- **Cohort descriptions:** We have improved the description of cohorts with specific characteristics and content to provide a clearer understanding of their selection and relevance to the study. The revised manuscript includes detailed information about the patient populations, treatments, demographics, and sample sources.
- **Linking molecular data analysis:** We have addressed the need for a clearer connection between the genome/gene/protein level analysis and the collagen/Tumor-infiltrating lymphocytes (TIL) analysis. Revised the Introduction and Discussion sections to emphasize the novel findings and how they relate to the main theme of the paper, i.e., the TIL-fiber relationship.
- **Added value of collagen integration:** We have demonstrated the enhanced predictive value of integrating collagen information into computational models, leading to improved C-indices and statistically significant results for survival outcomes.
- **Collagen fiber segmentation validation:** We have highlighted the results of visual assessment by two pathologists for validating collagen fiber segmentation in H&E-stained images.
- **Clinical relevance:** We have addressed the reviewer's request for clinically relevant molecular signatures, providing univariate analyses for molecular subtypes in endometrial carcinoma and HPV dependency in cervical carcinoma.

Reviewer #1 (Remarks to the Author)

“The study of Aggarwal et al. is investigating the role of collagen fibres and TILs in several gynecological malignancies. The study bears novelty as it integrates ECM elements into the computational perspective on malignancies. The study uses publicly available data sets from TCGA and hospital derived data sets. The cases were characterized regarding genetics and protein expression.”

“The study might be improved in these minor points”

- Comment 1: “There are many studies out for TIL evaluation with conventional digital pathology, but also computational models. It would be interesting to see more data about the "added value" achieved with the integration of the collagen part. This could be plotted e.g. with ROC curve comparisons”

Response 1: We appreciate the reviewer's inquiry regarding the added value achieved with the integration of collagen. We do acknowledge that previous approaches have explored computational modeling of TIL evaluation^{1,2}. However, our study differs from these previous approaches^{1,2} as it primarily focuses on investigating the relationship between the immune and collagen components within the tumor microenvironment of gynecologic cancers (ovarian, cervical, and endometrial), along with associating the TIL+collagen features with survival outcomes.

In our Supplementary Methods, we conducted a prognosis analysis using TIL, collagen, and TIL+collagen features. In response to your suggestion, we calculated the C-indices, which are more suitable for time-to-event and censored data, to evaluate the predictive performance of these models. Our findings indicate that combining TIL and collagen features resulted in higher C-indices, as well as statistically significant results (in terms of p-values, hazard ratios, and 95% confidence intervals), when compared to using TIL or collagen features alone for the validation cohorts (D1-D8) used in our study (Table 1).

Table 1. Prognosis analysis using TIL, collagen, and TIL+collagen components

	D1	D2	D3	D4	D5, D6, D7	D8
TIL component	c-index= 0.7 p=0.001, HR=4.3, 95% CI=2.05-9.03	c-index= 0.69 p=0.007, HR=2.96, 95% CI=1.52-5.73	c-index= 0.66 p=0.037, HR=3.38, 95% CI=1.34-8.51	c-index= 0.75 p=0.08, HR=3.11, 95% CI=0.727-13.3	c-index= 0.58 p=0.122, HR=1.79, 95% CI=0.9-3.54	c-index= 0.57 p=0.08, HR=3.07, 95% CI=0.9-9.85
Collagen component	c-index= 0.59 p=0.63, HR=1.2, 95%	c-index= 0.59 p=0.1, HR=1.74, 95%	c-index= 0.41 p=0.38, HR=0.676, 95%	c-index= 0.82 p=0.04, HR=3.1, 95%	c-index= 0.51 p=0.58, HR=0.823, 95% CI=0.42-1.62	c-index= 0.71 p=0.54, HR=1.59, 95% CI=0.422-5.96

	CI=0.57- 2.52	CI=0.91- 3.35	CI=0.63- 1.71	CI=1.02- 15.6		
TIL+collagen components	c-index= 0.72 p=0.0256, HR=2.54, 95% CI=1.21- 5.34	c-index= 0.72 p=0.006, HR=2.87, 95% CI=1.49- 5.53	c-index= 0.7 p=0.0213, HR=7.36, 95% CI=2.73- 19.8	c-index= 0.85 p=0.0259, HR=4.39, 95% CI=1.07-18	c-index= 0.65 p=0.0184, HR=2.72, 95% CI=1.36-5.42	c-index= 0.74 p=0.0434, HR=3.07, 95% CI=1.01-9.85

- Comment 2: “Was there an image-based groundtruth achieved for the collagen part, e.g. is the algorithm able to distinguish elastic fibers from collagen fibres. Did an overlay with histochemical (EVG) or immuno-histochemical images produce correct results.”

Response 2: We would like to emphasize that the ultimate validation of our collagen fiber segmentation method hinges upon its capacity to predict clinically relevant outcomes. Our current work demonstrates the feasibility of achieving this validation, where we show the association of features extracted from the quantitative characterization of the immune and collagen components within the tumor microenvironment of gynecologic cancers and survival outcomes.

We do acknowledge that not employing special staining techniques such as IHC or EVG is a limitation of our study, stemming from the unavailability of samples stained with these techniques. To evaluate the effectiveness of the method for segmenting collagen fibers from H&E WSIs, which was previously described in the literature³, we conducted a visual assessment involving two pathologists. They independently reviewed one randomly selected 3000x3000-pixel tile from 50 different patients randomly chosen from the training cohort (D0). The tiles were marked with collagen fibers captured by our approach. The two pathologists examined the tiles and categorized them into one of three categories (good, fair, or poor). For the collagen fiber segmentation, the first pathologist ranked 90% of the tiles as good or fair, while the second pathologist assigned a good or fair ranking to 92% of the tiles (Table 2).

Table 2. Quality check results of Collagen fiber segmentation

Task	Pathologist 1 (% of tiles belonging to good/fair category)	Pathologist 2 (% of tiles belonging to good/fair category)
Collagen fiber segmentation	90%	92%

We have included this limitation in the discussion section of the paper as follows:

We acknowledge that our study had certain limitations. One notable limitation was the absence of specialized staining techniques, such as immunohistochemistry (IHC) or Verhoeff-Van Gieson (EVG) staining, which would have enabled a quantitative analysis of collagen fiber segmentation. This limitation stemmed from the unavailability of IHC and EVG-stained whole slide images for the cohorts used in our study.

Regarding the reviewer's inquiry about the distinction between collagen fibers and elastic fibers, we acknowledge that this is a challenging aspect. To provide a thorough explanation, we sought guidance from pathologists to acquire deeper insights. While analyzing H&E-stained images to distinguish collagen fibers, we were meticulous in our approach, exclusively selecting what appeared to be the "thicker" fibers (specifically selected "dark line" collagen fibers in our Matlab code. "dark line" refers to one of the seven image structures that the model segmented each pixel of the whole slide image, namely, Flat, Slope, dark Blob, light Blob, dark line, light line, and Saddle). This selection criterion was based on the observation that collagen fibers generally exhibit a thicker and coarser appearance in contrast to the finer and more delicate nature of elastic fibers.

- Comment 3: “The tile format of 3000x3000 seems quite big in relation to the case numbers. Shouldn't the performance not be tested on larger validation sets? Please indicate the total number of tiles accessible for training and validation.”

Response 3: There seems to a confusion or misunderstanding regarding the use of tiles in our study, and we apologize for this. We extracted a set of non-overlapping 3000x3000-pixel tiles from each whole slide image because processing these enormous whole slide images in their entirety would be unrealistic due to computational limitations.

We then employ statistical measures (first-order statistics such as mean, minimum, maximum etc.) to effectively "summarize" the features extracted from all tiles of the whole slide image into a single feature vector. Therefore, the number of tiles used for training or validation does not directly impact our analysis. Ultimately, we distill the wealth of information contained in the tiles into a single vector per whole slide image.

Additionally, while different tile sizes could be considered, we have chosen a size of 3000x3000-pixel as it strikes a balance between capturing sufficient contextual information and managing the computational resources required for processing. This choice allows us to preserve essential details within the tiles while ensuring the feasibility of our approach in terms of computational power.

- Comment 4: “Clinically relevant molecular signatures/characteristics would be of interest. In endometrial carcinoma the molecular subtypes would be interesting, in cervix carcinoma the HPV dependency, in ovarian carcinoma the HRD status.”

Response 4:

Endometrial carcinoma:

We appreciate the reviewer’s comment. For the endometrial carcinoma validation cohorts (D3-D4) used in our study, univariate analysis using molecular subtypes (Cnhigh vs. Cnlow) was performed. Important to note that we didn’t have the molecular subtype clinical data for the endometrial carcinoma cohort (D7) used in our study. The results are shown in Table 3 and demonstrate that we didn’t find any association of the molecular subtypes with survival outcomes. We have added these results in the updated supplementary material as well (Supplementary Table 3).

Table 3. Univariate analysis using molecular subtypes for endometrial carcinoma validation cohorts (D3-D4)

Univariate analysis	D3	D4
Molecular subtypes (Cnhigh vs. Cnlow)	p=0.3, HR=1.58, 95% CI=0.64-3.89	p=0.9, HR=1.06, 95% CI=0.2-5.4

Cervix carcinoma:

For the cervical carcinoma validation cohorts (D1-D2) used in our study, univariate analysis using HPV dependency data (HPV+ vs. HPV-) was performed. Important to note that we didn't have the HPV dependency data for the cervical carcinoma cohort (D6) used in our study. The results are shown in Table 4 and demonstrate that we didn't find any association of the HPV dependency data with survival outcomes. We have added these results in the updated supplementary material as well (Supplementary Table 3).

Table 4. Univariate analysis using HPV dependency data for cervical carcinoma validation cohorts (D1-D2)

Univariate analysis	D1	D2
HPV dependency data (HPV+ vs. HPV-)	p=0.38, HR=1.18, 95% CI=1.1-6.9	p=0.7, HR=1.47, 95% CI=1.1-7.73

Ovarian carcinoma:

Unfortunately, we didn't have the HRD status data available for the ovarian carcinoma validation cohorts used in our study (D5 and D8).

Reviewer #2 (Remarks to the Author)

“The authors suggest a hypothesis that a combination of TIL TME composition and collagen fiber structure plays a role in determining patient outcome in several gynecological cancers. Across several cohorts they show how disorganized fibers correspond to higher degree of immune infiltration, while regular fiber structure to lower immune infiltration and worse patient outcomes. TME analysis is followed by genome, gene expression and protein level analysis in a subset of cohorts.

I find the idea of analyzing TME fibers together with the TIL composition very interesting and worth investigating. While their correspondence is well presented and their effect in patient groups is demonstrated, I find several drawbacks of the analysis and result presentation in the manuscript. Additionally, the molecular data analysis lacks any link to the WSI analysis and is presented in a way that is hard to follow or comprehend with relationship to the main finding of the paper (TIL-fiber relationship).”

Response: We sincerely appreciate your thoughtful and constructive feedback on our manuscript. Your comments are invaluable in helping us refine and improve our work.

Major points

Comment 1: “Across the paper, the cohorts, diseases, datasets are not well described. The results section should start by describing D0-D8, with their specific characteristics and content. Otherwise, the selection of cohorts for different parts of the analysis seems arbitrary and is hard to understand. Why, for example, D0 is the training cohort?”

Response 1: We apologize for not having included a complete description of the different cohorts used in our study. We have updated the starting of the results section in the updated manuscript draft by adding more information about the cohorts (D0-D8) with their specific characteristics and content as follows:

Patient populations for the study

We analyzed 493 patients diagnosed with HGSOC (n=139), CSCC (n=269), or EC (n=85) from various sites, including The Cancer Genome Atlas (TCGA, n=357)⁴, University Hospitals (UH, n=58), Cleveland Clinic (CCF, n=48), and Memorial Sloan Kettering Cancer Center (MSKCC, n=30). For the patients from TCGA diagnosed with HGSOC (D0, n=95), tissue samples were obtained through cytoreductive surgery before the initiation of chemotherapy. Similarly, for the patients from TCGA diagnosed with CSCC (D1 and D2, n=262), tissue samples were obtained through

cytoreductive surgery before the initiation of chemotherapy or radiotherapy. All patients from UH (D3 and D4, n=58) underwent surgery (hysterectomy with bilateral salpingo-oophorectomy), the standard treatment for EC. The patients in these cohorts who were considered as intermediate- or high-risk for recurrence received chemotherapy following surgery (D3, n=32). The CCF cohorts (D5, D6, and D7) included 48 patients treated with immunotherapy agents, including pembrolizumab, nivolumab±ipilimumab, and avelumab, all in the recurrent setting (D5 for HGSOC, D6 for CSCC, and D7 for EC). All patients from MSKCC (D8, n=30) were diagnosed with HGSOC and underwent primary debulking surgery.

The training cohort (D0) consisted of 95 patients from TCGA diagnosed with HGSOC, while the validation cohorts (D1-D8) comprised a total of 398 patients diagnosed with HGSOC, CSCC, or EC from various sites. The cohorts (D0-D8) comprised 71 patients with stage IV disease, 153 with stage III, 68 with stage II, 192 with stage I, and 9 with an unknown stage. All H&E imaging for patients from these sites was carried out before the initiation of therapy (chemotherapy, radiotherapy, or immunotherapy). The summary of cohort characteristics, censor statistics, treatment strategy, demographics, and included samples can be found in Fig. 2, Table 1, and Methods.

It is important to note that we utilized the D0 cohort, which represents the TCGA ovarian cancer cohort, for training. Typically, TCGA cohorts are known for their heterogeneity as they consist of data from various institutions.

Comment 2: "I see no link between genome/gene/protein level analysis with the collagen/TIL analysis. These seem like two separate stories. What is the novelty in the findings in the molecular data?"

Response 2: We appreciate the reviewer's comment and have carefully considered the feedback. In response, we have made revisions to the manuscript, particularly in the Introduction and Discussion sections. The revisions are aimed at highlighting the uniqueness of our findings and describing a clearer connection between the genome/gene/protein level analysis and the collagen/TIL analysis.

The second objective of our study was to conduct molecular investigations within the TME of gynecologic cancers. The aim was to identify genomic changes that correlate with alterations in histological patterns, specifically distinguishing between high-risk and low-risk patients. A key innovation of our study is the investigation of whether the collagen/TIL relationship observed through CollaTIL extends to the molecular level. By integrating genomic analysis, we seek to identify relevant biological pathways and gene signatures that are associated with the observed collagen-TIL interactions. This innovative approach allows us to uncover the afore-mentioned biological pathways and gene signatures that underlie both favorable and unfavorable clinical outcomes in patients with HGSOC and CSCC.

In essence, our research endeavors to establish a comprehensive study from the morphological observations within the TME to the molecular-level correlations. By these analyses, we provide a more comprehensive understanding of the intricate interplay between immune-mediated collagen disruption, corresponding molecular changes, and its impact on patient outcomes.

Comment 3: "What is the link between the "Amino acid and Macrophage gene signatures" and TILs/collagen fiber composition?"

Response 3: We appreciate the reviewer's question and recognize the significance of elucidating this connection. In response, we have made corresponding updates to the manuscript. The key points summarizing the association between "Amino acid and Macrophage gene signatures" and Tumor-Infiltrating Lymphocytes (TILs) and collagen fiber composition are outlined below:

HGSOC Patients (D0): Among the identified genes in HGSOC patients (D0), we found a subset of gene signatures related to amino acid metabolism and macrophages that were significantly associated with a higher risk score predicted by CollaTIL⁵.

Amino acid metabolism plays a pivotal role in macrophage generation and polarization, influencing their phenotypes (M1 or M2)⁶. M1 macrophages can contribute to extracellular matrix (ECM) degradation and promote an anti-tumor immune response, whereas M2 macrophages can facilitate ECM remodeling that promotes tumor progression⁷.

These findings suggest the possibility of enhanced protumorigenic M2 macrophages in high-risk HGSOC patients, where the presence of amino acid gene signature may indicate weakened immune activity and a positive correlation with angiogenesis-associated genes^{8,9}.

CSCC Patients (D1-D2): In CSCC patients, we identified gene signatures related to the tricarboxylic acid (TCA) cycle, the downregulation of which may reflect shifts in metabolic pathways. These TCA-cycle genes are part of a gene signature linked to predicting cancer prognosis in CSCC patients. Although their specific role in ECM degradation, particularly collagen, remains unclear, their downregulation can influence cellular metabolism, energy production, and redox balance, which in turn could impact ECM remodeling processes. Our analysis also identified specific genes significantly correlated with OS for CSCC and HGSOC patients, emphasizing their potential as prognostic factors. Furthermore, we highlighted genes strongly associated with risk scores predicted by CollaTIL, suggesting their significance in disease progression and prognosis¹⁰.

Our study provides evidence of a molecular link between the "Amino acid and Macrophage gene signatures" and TILs/collagen fiber composition. These findings underscore the intricate interplay between immune-mediated processes, cellular metabolism, ECM remodeling, and their implications for patient outcomes in gynecologic malignancies. We hope this clarification helps illuminate the connection between these aspects of our research.

Comment 4: "Is CollaTIL a tool? I didn't find any link to a code repository. In that case the manuscript presents an analysis, it shouldn't have a name."

Response 4: Yes, CollaTIL is a tool. The code is available at this link: https://github.com/arp95/collatil_biomarker_gyn

Comment 5: "The molecular data analysis is unordered and hard to follow. The results should be selected in a way to build a coherent story that links the following parts together."

Response 5: We appreciate the reviewer's feedback, and we understand the importance of presenting our results in a coherent and logical sequence to build a compelling narrative. In our manuscript, we aimed to systematically structure our molecular data analysis to elucidate the associations and prognostic implications surrounding CollaTIL and the presence of mutated genes in HGSOC and CSCC patients (D0-D2). Our approach can be summarized as follows:

1. Identification of Mutations and Prognostic Implications: Through our analysis, we explored the relationship between frequently somatic mutations and patient outcomes. We initiated the analysis by systematically categorizing a subset of patients from the D0-D2 cohorts. This categorization was based on the availability of mutation annotations and

H&E WSIs. Patients were divided into two distinct groups: those harboring gene mutations and those exhibiting wild-type gene status. Specifically, we identified the association of nonsynonymous somatic mutations in PIK3CA with the risk of recurrence or death in CSCC patients (D1, D2). These findings helped reveal that PIK3CA mutations were linked to a diminished risk of recurrence or death in a specific subset of patients.

2. Gene Signature and Differential Expression Analysis: We further aimed to identify genes that could have impact associated with CollaTIL in CSCC patients (D1-D2). To achieve this, we initially identified genes whose expressions correlated significantly with estimated risk scores, as predicted by CollaTIL. Subsequently, we performed an analysis of differentially expressed genes to pinpoint significant candidates among patients at high and low-risk, using the threshold of false discovery rate (FDR) <0.05 and \log_2 -fold change >1 .

3. Single-Sample Gene Set Enrichment Analysis (ssGSEA): To delve deeper into the molecular implications, we utilized ssGSEA methodology to estimate the activity index of identified gene signatures¹¹. This analysis revealed distinct expression patterns within high- and low-risk groups of CSCC patients, highlighting the downregulation of genes associated with the tricarboxylic acid (TCA) cycle in high-risk patients.

4. Prognostic Genes: We concluded the molecular analysis by conducting a comprehensive exploration of the association between the top identified genes in CSCC and HGSOC with patient overall survival. Notably, we identified specific genes, including APOBEC3H, KIRREL1, and FAM166B, that were significantly correlated with overall survival, suggesting their potential as prognostic factors.

5. Protein Expression and Genetic Potential Analysis: Lastly, we explored protein expression modulation and assessed genetic potential associated with high-risk and low-risk groups identified using CollaTIL. We observed significant differences in protein expression of genes related to macrophages and Amino acids in HGSOC patients. Furthermore, we utilized CRISPR knockout data to assess the impact of highly altered genes in HGSOC cell lines, to highlight their functional potentials¹².

In our revised manuscript, we ensure that these aspects of our molecular data analysis are presented in a more ordered and cohesive manner, strengthening the narrative flow and facilitating a clearer understanding of the relationships between CollaTIL, mutated genes, and molecular insights.

Detailed points

Comment 6: “L78-80 “previous findings..” lack citation”

Response 6: We thank the reviewer for pointing this out. We have added the required citations for this line in the updated manuscript draft.

Comment 7: “L125-126 An existing pretrained deep learning model was used to segment the epithelial and stromal regions on these tiles. The authors should explicitly mention it here that these results were additionally validated by pathologists. I only found this information while scrolling through the Supplementary Material.”

Response 7: We totally agree with the reviewer. Consequently, we have added the visual assessment part by the pathologists in the Results section of the updated manuscript draft as follows:

To assess the performance of these models, we conducted a visual assessment with two pathologists. They independently reviewed one randomly selected 3000x3000-pixel tile from 50 different patients randomly chosen from the training cohort (D0). The two pathologists examined the tiles and categorized them into one of three categories (good, fair, or poor). For the nuclei segmentation, both pathologists unanimously ranked 100% of the tiles as either good or fair. However, for the epithelium/stroma segmentation, the first pathologist ranked 90% of the tiles as good or fair, while the second pathologist assigned a good or fair ranking to 94% of the tiles (Fig. 3a, Methods, and Supplementary Table 1).

Comment 8: “L133 ‘derivative-of-Gaussian based model’ is not self-explanatory. Can you explain what is it and how is it used to “capture the fiber orientations by detecting linear structures within these regions.”

Response 8: We apologize for not having explained this in the paper. We have added this explanation in the updated supplementary draft as follows:

Steps involved in extracting features from collagen component

The steps involved in extracting features from collagen component are as follows:

- **Detecting collagen fibers**
- **Find the orientation of the detected collagen fibers**
- **Discretize the orientations in 18 bins (0 to 17)**
- **Then tumor neighborhoods of nine different sizes (200x200-pixel, 250x250-pixel, 300x300-pixel, 350x350-pixel, 400x400-pixel, 450x450-pixel, 500x500-pixel, 550x550-pixel, 600x600-pixel) move across the extracted tiles of whole slide images in a sliding window way, calculate the disorder of collagen fiber orientations by constructing an orientation co-occurrence matrix within each tumor neighborhood**
- **Quantitative measurement of the disorder in collagen fiber orientations was measured using the orientation co-occurrence matrix using entropy theory**

Step 1: Detecting collagen fibers using derivative of Gaussian model

- **We used the derivative of Gaussian based model in Matlab to detect the linear structures in the image, which are the prevalent morphology phenotype exhibited by collagen fibers on H&E images.**

Step 2: Orientation of detected collagen fibers (regionprops function in Matlab software)

- **We use the collagen fibers that are detected by the derivative-of-Gaussian model. Essentially for each pixel we compute the x-gradient, y-gradient**

and that gives us the orientation (Orientation (θ) = atan2(y-gradient, x-gradient))

- *For each collagen fiber, we create the histogram of orientations for each pixel as computed above and find the orientation that occurs more frequently that is considered as the orientation for that collagen fiber.*

Step 3: Discretize the orientations in 18 bins from 0 to 17

Step 4: Orientation co-occurrence matrix

Divide each tile extracted from the whole slide image in tumor neighborhoods of nine different sizes (200x200-pixel, 250x250-pixel and so on). Within each neighborhood, compute the orientation co-occurrence matrix where each row and column of the matrix denote the frequency of orientations having value row or column.

Step 5: Quantitative measurement of the disorder in collagen fiber orientations

After obtaining the orientation co-occurrence matrix for each neighborhood, we essentially use this matrix to compute the disorder in collagen fiber orientations using entropy theory.

Comment 9: “L183 What kind of molecular data? Why wouldn’t you make this comparison in cohorts D1-D8?”

Response 9: We apologize for any confusion regarding this matter. In response to the reviewer's concern, we have included a statement in the Methods section of our manuscript to elucidate the limitations in data availability for cohorts D3-D8 and the rationale guiding our emphasis on cohorts D0-D2.

A more detailed clarification regarding our decision to exclusively use molecular data for cohorts D0-D2 is provided below:

Our choice to focus on molecular data for cohorts D0-D2 stems from constraints in data availability. We acquired molecular data, including mutation annotations, gene expression data (raw read counts for differential expression analysis, and FPKM values for other analyses), and protein expression, from the TCGA datasets. Unfortunately, these molecular datasets were inaccessible for cohorts D3-D8, leading to their exclusion from the genomics analysis.

Comment 10: “Fig 5. Why wouldn’t you combine these plots? Cohorts can be marked with different colors in a single plot.”

Response 10: The reviewer is right. We have combined plots 1-3 and plots 4-6 and have updated the figure (**Fig. 5**) in the manuscript draft attached to this submission.

Comment 11: “Fig 6. What is “Signature index”?”

Response 11: For clarification, we have updated the text in the Supplementary Methods of the revised manuscript. In Single Sample Gene Set Enrichment Analysis (ssGSEA) enrichment analysis tools, the term "signature index" typically refers to a numerical score or metric that quantifies the extent to which a particular gene expression signature or gene set is enriched or depleted in a single sample or specimen. It summarizes certain characteristics or properties of data¹¹. In ssGSEA, the signature index assigned to a sample, indicating how closely its gene expression profile aligns with gene expression pattern of a particular gene set or signature. It quantifies the degree of enrichment (positive index) or depletion (negative index) of genes within the signature in the sample.

We used signature index values to gain information about the pathways that are activated or suppressed in individual samples, helping us understand the underlying biology of the high-risk and low-risk cohorts in the context of the identified gene sets.

Comment 12: “L204 What are “risk scores” how are they estimated and what are “race-specific architectural-based models of TME”? There is no citation here either.”

Response 12: We apologize for the lack of clarity in describing how the risk scores are calculated. The CollaTIL method accomplishes two main objectives: 1) determining features related to overall survival and progression-free survival and 2) classifying patients as low or high risk. It's essential to note that these two aspects are highly correlated as the features associated with overall survival or progression-free survival play a crucial role in identifying a patient's risk level.

More specifically, CollaTIL comprises 34 features, with 27 from the collagen component and 7 from the immune component. A Cox proportional hazards regression model¹³ in conjunction with elastic net penalty¹⁴ was used to identify the top CollaTIL features (i.e., those that are most associated with patient outcome) in the training set (D0) along with their coefficients (which indicate the importance of each feature) for overall survival. Risk scores were computed for each patient in D0 as a linear combination of the top feature values and their respective coefficients. Finally, the mean value of all risk scores of patients in D0 was computed and used as a cutoff to discriminate among patients as low or high-risk. A patient whose risk score value is higher than the mean is considered high risk, and a patient with a risk score value lower than the mean is considered low risk. Our CollaTIL signature allowed to distinguish between high and low-risk patients across multiple disease types (ovarian, cervical, and endometrial cancers) and treatment indications (chemotherapy, radiotherapy, and immunotherapy).

We deeply apologize for the confusion regarding “race-specific architectural-based models of TME”. This was in fact a typo in the submitted manuscript, and we have updated this line in the main text.

Comment 13: “L207 Abbreviation FDR is not explained.”

Response 13: We thank the reviewer for this remark and have made the necessary clarification in the text. FDR stands for False Discovery Rate. In our multiple hypothesis testing of gene expression data, the False Discovery Rate addresses a specific challenge that traditional p-values or family-wise error rates (FWER) do not fully account for the control of false positives when testing multiple hypotheses simultaneously¹⁵.

Comment 14: “L209 What are the “two distinct set of genes”? Are these different sets of genes for cohort D1 and D2 (in that case add “respectively” at the end of the sentence). In any case those “sets of genes” need to be explained.”

Response 14: We thank the reviewer for this remark. We have made the necessary change to the text. For clarification we have added the list of the detailed gene sets in the **Supplementary Table 6**.

Comment 15: “L213 “activity index” is not self-explanatory, what is its meaning?”

Response 15: We thank the reviewer for this remark. In the context of single-sample gene set enrichment analysis (ssGSEA), the "activity index", or “Signature index” refers to a metric or score that quantifies the activity or expression level of a specific gene set or pathway within an individual sample or observation. It is used to assess the relative activity or enrichment of a particular biological process, pathway, or gene set in a given sample¹¹.

What the activity index represents: For each sample, ssGSEA calculates an activity index for one or more predefined gene sets. The activity index is determined by assessing the collective expression levels of the genes within the gene set. Several statistical methods can be used for this calculation, but one common approach is to compute a weighted sum or enrichment score that reflects the degree to which the genes in the set are coordinately upregulated or downregulated in the sample. The resulting activity index provides a quantitative measure of the degree to which a specific biological process or pathway is active or enriched in the given sample. A higher activity index indicates a higher level of activity or enrichment of the gene set in the sample, while a lower index suggests lower activity^{11,16}.

In cancer research, ssGSEA can be applied to assess the activation of specific signaling pathways or immune-related gene sets in tumor samples. This information can be valuable for understanding disease mechanisms, identifying biomarkers, or stratifying patients based on the activity of specific pathways.

Comment 16: “L223 SSGSEA, or ssGSEA (few lines before)?”

Response 16: We thank the reviewer for this remark and have made the necessary change to the text.

Comment 17: “L227 What do you mean by “top identified genes”? Based on what ranking are these top genes?”

Response 17: We appreciate the reviewer seeking clarification. We have addressed this concern by making a necessary modification to the text, specifically removing the word "top" to eliminate any ambiguity. For clarification we have added the list of the detailed gene sets in the **Supplementary Table 6**.

Comment 18: "L237-245 What do these genes mean, are they related to TILs or fibers?"

Response 18: Thank you for the comment. We have updated the discussion section of the main text for clarification.

Comment 19: "L246 "highly altered genes" – do you mean here expression, protein level or mutations?"

Response 19: We thank the reviewer for requesting this clarification. We have updated the text for clarification. We meant altered expression of gene or protein in comparison to normal tissue cohort.

Comment 20: "L246-255 What is the reason to verify impact on cellular growth here? I don't see what is the logical thread here."

Response 20: We identified the altered expression of indicated gene or protein in HGSOC as compared to the normal tissue, to evaluate functional relevance of these genes in HGSOC, we leverage the databases of Broad's Achilles and Sanger's SCORE projects, which evaluate gene effects by CRISPR knockout in various HGSOC cell lines and estimate impact of these genes on cellular growth^{17,18}.

Comment 21: "Fig. 8. What is "Chronos"?"

Response 21: Our apology for oversight, Chronos is a mechanistic framework designed to utilize the detailed behavior of pooled CRISPR experiments to improve inference of gene essentiality^{12,19}, where "gene essentiality" refers to the significance or necessity of a particular gene for the survival, growth, or normal functioning of a cell or organism.

Comment 22: "L297 'low-risk patients identified by CollaTIL' – does your method identify the patients or rather features related to the OS and PFS?"

Response 22: The method accomplishes two main objectives: 1) determining features related to overall survival and progression-free survival and 2) classifying patients as low or high risk. It's essential to note that these two aspects are highly correlated as the features associated with overall survival or progression-free survival play a crucial role in identifying a patient's risk level.

More specifically, CollaTIL comprises 34 features, with 27 from the collagen component and 7 from the immune component. A Cox proportional hazards regression model¹³ in conjunction with elastic net penalty¹⁴ was used to identify the top CollaTIL features (i.e., those that are most associated with patient outcome) in the training set (D0) along with

their coefficients (which indicate the importance of each feature) for overall survival. Risk scores were computed for each patient in D0 as a linear combination of the top feature values and their respective coefficients. Finally, the mean value of all risk scores of patients in D0 was computed and used as a cutoff to discriminate among patients as low or high-risk. A patient whose risk score value is higher than the mean is considered high risk, and a patient with a risk score value lower than the mean is considered low risk. Our CollaTIL signature allowed to distinguish between high and low-risk patients across multiple disease types (ovarian, cervical, and endometrial cancers) and treatment indications (chemotherapy, radiotherapy, and immunotherapy).

Comment 23: "L306 'Amino acid and Macrophage gene signatures..' how are these genes related to the results of WSI analysis? Or is it 2 separate analyses?"

Response 23: We thank the reviewer's question and recognize the significance of elucidating this connection. In response, we have made corresponding updates to the manuscript. The key points summarizing the association between "Amino acid and Macrophage gene signatures" and Tumor-Infiltrating Lymphocytes (TILs) and collagen fiber composition are outlined below:

HGSOC Patients (D0): Among the identified genes in HGSOC patients (D0), we found a subset of gene signatures related to amino acid metabolism and macrophages that were significantly associated with a higher risk score predicted by CollaTIL⁵.

Amino acid metabolism plays a pivotal role in macrophage generation and polarization, influencing their phenotypes (M1 or M2)⁶. M1 macrophages can contribute to extracellular matrix (ECM) degradation and promote an anti-tumor immune response, whereas M2 macrophages can facilitate ECM remodeling that promotes tumor progression⁷.

These findings suggest the possibility of enhanced protumorigenic M2 macrophages in high-risk HGSOC patients, where the presence of amino acid gene signature may indicate weakened immune activity and a positive correlation with angiogenesis-associated genes^{8,9}.

CSCC Patients (D1-D2): In CSCC patients, we identified gene signatures related to the tricarboxylic acid (TCA) cycle, the downregulation of which may reflect shifts in metabolic pathways. These TCA-cycle genes are part of a gene signature linked to predicting cancer prognosis in CSCC patients. Although their specific role in ECM degradation, particularly collagen, remains unclear, their downregulation can influence cellular metabolism, energy production, and redox balance, which in turn could impact ECM remodeling processes. Our analysis also identified specific genes significantly correlated with OS for CSCC and HGSOC patients, emphasizing their potential as prognostic factors. Furthermore, we highlighted genes strongly associated with risk scores predicted by CollaTIL, suggesting their significance in disease progression and prognosis¹⁰.

Our study provides evidence of a molecular link between the "Amino acid and Macrophage gene signatures" and TILs/collagen fiber composition. These findings underscore the intricate interplay between immune-mediated processes, cellular

metabolism, ECM remodeling, and their implications for patient outcomes in gynecologic malignancies. We hope this clarification helps illuminate the connection between these aspects of our research.

Comment 24: “L381 Why was the model trained on “oral cavity tissue microarray images”? What was the test set performance?”

Response 24: It is important to note that we did not perform any training of the models for the preprocessing steps, such as epithelium/stroma and nuclei segmentation. We utilized pretrained models from the literature for our analysis²⁰. For epithelium/stroma segmentation, we employed a model trained on oral cavity tissue microarray images. The reason we went ahead with this model is because the epithelium and stroma regions look similar across organs, so it should not be a problem to apply a model trained on oral cavity tissue microarray images to gynecologic cancer (ovarian, cervical, endometrial) images.

Since ground truth data for the preprocessing steps was unavailable, we conducted a visual assessment by two pathologists on the output generated by the pretrained model for epithelium/stroma segmentation task. They independently reviewed one randomly selected 3000x3000-pixel tile from 50 different patients randomly chosen from the training cohort (D0). The two pathologists examined the tiles and categorized them into one of three categories (good, fair, or poor). For the epithelium/stroma segmentation, the first pathologist ranked 90% of the tiles as good or fair, while the second pathologist assigned a good or fair ranking to 94% of the tiles (Table 5).

Table 5. Quality check results of Epithelium/Stroma segmentation

Task	Pathologist 1 (% of tiles belonging to good/fair category)	Pathologist 2 (% of tiles belonging to good/fair category)
Epithelium/Stroma segmentation	90%	94%

Comment 25: “L385 How was the threshold “fine-tuned”?”

Response 25: We deeply apologize for the confusion here. This was in fact a typo in the submitted manuscript. There was no fine-tuning of the threshold performed but a specified value of the threshold was used. The binary masks obtained using this specified threshold for epithelium/stroma task were then visually assessed by pathologists. The results of the visual assessment by pathologists are shown in Table 5. We have removed this line from the updated manuscript draft.

Comment 26: “L388 How were the regions selected? Manually?”

Response 26: No, the regions were not selected manually. Our analysis focused on the epithelial regions, which were automatically segmented using a state-of-the-art algorithm²⁰. To avoid confusion, we have removed this line from the updated manuscript draft as there are not manually selected epithelial regions and instead the binary mask output from epithelium/stroma segmentation preprocessing step were used for subsequent analysis.

Comment 27: “L406 Why not use HoverNet cell classification results to identify TILs? How accurate is the classification method proposed by the authors?”

Response 27: The reviewer raises a very interesting question. We have added the explanation of why we didn't use HoverNet for TIL classification in the updated supplementary draft as follows:

We conducted an experiment to determine which TIL classification model we should use for the present work: SVM²¹ or Hover-Net²². For this, we randomly selected 80 tiles of size 3000x3000-pixel from the training cohort (D0). We applied both approaches to the tiles and asked two pathologists to select the model that provided the best TIL detection. One of the pathologists stated that the SVM model performed better in 52% of the tiles, Hover-Net in 28%, and there was not a significant difference in the remaining 20%. The second pathologist noted that the SVM was better in 75% of the cases, Hover-Net in 19%, and no significant difference was observed in 6% of the cases. We asked the pathologists for their overall impression of the models, and both of them agreed that while Hover-Net was very precise, it had lower recall. For this reason, we chose to use the SVM model.

Comment 28: “L423 Why bins are from 0 to 17? What do they correspond to and why 17?”

Response 28: Since orientation values fall within the range of 0 to 180 degrees, we chose this discretization as a hyperparameter setting it to 18 bins. Each bin represented

orientations within the ranges of 0-10, 10-20, 20-30, 30-40, and so on. Increasing the number of bins would have a computational impact on the algorithm, increasing the time required for feature extraction from the collagen component.

Comment 29: "L451 I don't see which 7 features these are in the Supplementary Table 1. There are 6 TIL-related features in the table."

Response 29: We apologize for the confusion. The CollaTIL signature consisted of 27 features from collagen component and 7 features from TIL component. Supplementary Table 1 lists the top features (6 from TIL component and 8 from collagen component) selected by the Cox regression model trained on the D0 cohort to predict overall survival.

The **Table 6** below shows the 7 features used from the TIL component for training the Cox regression model.

We have added the list of features from the TIL component and collagen component in the updated supplementary draft (Supplementary Table 7 and Supplementary Table 8).

Table 6. Features from TIL component

Feature index	Feature description
1	Ratio of non-TILs density to the surrounding (20 microns proximity) TILs in the epithelium compartment
2	Number of epithelial TIL clusters surrounding (20 microns proximity) a non-TIL cluster in the epithelium compartment
3	Presence percentage (ratio of present clusters to total number of clusters) of stromal non-TIL clusters being around another non-TIL cluster in the stromal compartment
4	Intersected area of clusters of epithelial TILs and non-TILs in invasive tumor front compartment
5	Minimum area of stromal TIL clusters in invasive tumor front compartment
6	Range of area of epithelial non-TIL clusters in invasive tumor front compartment
7	Range of density of TIL clusters to the surrounding non-TIL ones in stroma

Comment 30: “L464 ‘CollaTIL-defined low- and high-risk patients’ – again are these patients defined by CollaTIL? How?”

Response 30: We apologize for the lack of clarity in describing how the low and high-risk patients are identified. The CollaTIL method accomplishes two main objectives: 1) determining features related to overall survival and progression-free survival and 2) classifying patients as low or high risk. It's essential to note that these two aspects are highly correlated as the features associated with overall survival or progression-free survival play a crucial role in identifying a patient's risk level.

More specifically, CollaTIL comprises 34 features, with 27 from the collagen component and 7 from the immune component. A Cox proportional hazards regression model¹³ in conjunction with elastic net penalty¹⁴ was used to identify the top CollaTIL features (i.e., those that are most associated with patient outcome) in the training set (D0) along with their coefficients (which indicate the importance of each feature) for overall survival. Risk scores were computed for each patient in D0 as a linear combination of the top feature values and their respective coefficients. Finally, the mean value of all risk scores of patients in D0 was computed and used as a cutoff to discriminate among patients as low or high-risk. A patient whose risk score value is higher than the mean is considered high risk, and a patient with a risk score value lower than the mean is considered low risk.

Comment 31: “L506 In my opinion, with the publication the datasets should be public. Otherwise, we cannot warranty reproducibility of this research.”

Response 31: We totally understand the reviewer point. In fact, all H&E WSI and genomic data from TCGA cohorts (D0-D2) is publicly available from the TCGA Research Network (<http://cancergenome.nih.gov/>) and The Cancer Imaging Archive (<https://www.cancerimagingarchive.net/>). However, we wish to clarify that we do not possess ownership of the remaining datasets (D3-D8) as they were provided by different institutions (i.e., University Hospitals, Cleveland Clinic, Memorial Sloan Kettering Cancer Center) under license / by permission. Researchers who seek access to these datasets are kindly requested to initiate contact with the respective institution in order to secure the necessary permissions.

References

1. Azarianpour, S. *et al.* Computational image features of immune architecture is associated with clinical benefit and survival in gynecological cancers across treatment modalities. *J Immunother Cancer* **10**, (2022).
2. Corredor, G. *et al.* Spatial architecture and arrangement of tumor-infiltrating lymphocytes for predicting likelihood of recurrence in early-stage non-small cell lung cancer. *Clinical Cancer Research* **25**, 1526–1534 (2019).
3. Li, H. *et al.* Collagen fiber orientation disorder from H&E images is prognostic for early stage breast cancer: clinical trial validation. *NPJ Breast Cancer* **7**, (2021).
4. Tomczak, K., Czerwińska, P. & Wiznerowicz, M. The Cancer Genome Atlas (TCGA): An immeasurable source of knowledge. *Wspolczesna Onkologia* vol. 1A A68–A77 Preprint at <https://doi.org/10.5114/wo.2014.47136> (2015).
5. Wei, Z., Liu, X., Cheng, C., Yu, W. & Yi, P. Metabolism of Amino Acids in Cancer. *Frontiers in Cell and Developmental Biology* vol. 8 Preprint at <https://doi.org/10.3389/fcell.2020.603837> (2021).
6. Kieler, M., Hofmann, M. & Schabbauer, G. More than just protein building blocks: how amino acids and related metabolic pathways fuel macrophage polarization. *FEBS Journal* vol. 288 3694–3714 Preprint at <https://doi.org/10.1111/febs.15715> (2021).
7. Boutilier, A. J. & Elsawa, S. F. Macrophage polarization states in the tumor microenvironment. *International Journal of Molecular Sciences* vol. 22 Preprint at <https://doi.org/10.3390/ijms22136995> (2021).
8. El-Arabey, A. A. *et al.* Revisiting macrophages in ovarian cancer microenvironment: development, function and interaction. *Medical Oncology* vol. 40 Preprint at <https://doi.org/10.1007/s12032-023-01987-x> (2023).
9. Malekghasemi, S. *et al.* Tumor-associated macrophages: Protumoral macrophages in inflammatory tumor microenvironment. *Advanced Pharmaceutical Bulletin* vol. 10 556–565 Preprint at <https://doi.org/10.34172/apb.2020.066> (2020).
10. Anderson, N. M., Mucka, P., Kern, J. G. & Feng, H. The emerging role and targetability of the TCA cycle in cancer metabolism. *Protein and Cell* vol. 9 216–237 Preprint at <https://doi.org/10.1007/s13238-017-0451-1> (2018).
11. Barbie, D. A. *et al.* Systematic RNA interference reveals that oncogenic KRAS-driven cancers require TBK1. *Nature* **462**, 108–112 (2009).
12. Dempster, J. M. *et al.* Agreement between two large pan-cancer CRISPR-Cas9 gene dependency data sets. *Nat Commun* **10**, (2019).
13. Cox, D. R. Regression Models and Life-Tables. *Journal of the Royal Statistical Society: Series B (Methodological)* **34**, 187–202 (1972).
14. Wu, Y. Elastic net for Cox’s proportional hazards model with a solution path algorithm. *Stat Sin* **22**, 271–294 (2012).
15. Benjamini, Y. & Hochberg, Y. *Controlling the False Discovery Rate: A Practical and Powerful Approach to Multiple Testing*. Source: *Journal of the Royal Statistical Society: Series B (Methodological)* vol. 57 (1995).
16. Subramanian, A. *et al.* *Gene set enrichment analysis: A knowledge-based approach for interpreting genome-wide expression profiles*. www.pnas.org/cgi/doi/10.1073/pnas.0506580102 (2005).

17. Dwane, L. *et al.* Project Score database: A resource for investigating cancer cell dependencies and prioritizing therapeutic targets. *Nucleic Acids Res* **49**, D1365–D1372 (2021).
18. Cowley, G. S. *et al.* Parallel genome-scale loss of function screens in 216 cancer cell lines for the identification of context-specific genetic dependencies. *Sci Data* **1**, (2014).
19. Dempster, J. M. *et al.* Chronos: a cell population dynamics model of CRISPR experiments that improves inference of gene fitness effects. *Genome Biol* **22**, (2021).
20. Wu, Y. *et al.* A machine learning model for separating epithelial and stromal regions in oral cavity squamous cell carcinomas using H&E-stained histology images: A multi-center, retrospective study. *Oral Oncol* **131**, (2022).
21. Romero Castro, E. *et al.* A watershed and feature-based approach for automated detection of lymphocytes on lung cancer images. in 26 (SPIE-Intl Soc Optical Eng, 2018). doi:10.1117/12.2293147.
22. Graham, S. *et al.* Hover-Net: Simultaneous segmentation and classification of nuclei in multi-tissue histology images. *Med Image Anal* **58**, (2019).

REVIEWERS' COMMENTS:

Reviewer #1 (Remarks to the Author):

The study of Aggarwal et al. gained clarity during the first revisions. The implementation of pathologist is highly appreciated. However, there are still some minor points to be considered

- Figure 1 would profit from examples of real life heatmap CollaTil images aside the schematic view
- Figure 8 about cell lines seems a little bit out of topic as the complexity of the TME is not reflected by cell lines only. This information could go into supplemental.
- Instead of this supplemental figure1 could enter the main manuscript.
- The same could be discussed with tables. From a personal point of view, feature tables should be with the main text, but the cohort description could enter the supplemental.

These are only minor suggestions to a more convenient presentation of the many data generated by this study.

Reviewer #2 (Remarks to the Author):

The manuscript improve substantially and I recommend it for publication.

Wallace H. Coulter Department of
Biomedical Engineering

Paper number: COMMSMED-23-0509B

Communications Medicine

Title: Computational pathology identifies immune-mediated collagen disruption to predict clinical outcomes in gynecologic malignancies

Dear Reviewers,

Please find enclosed a revised version of our manuscript “**Computational pathology identifies immune-mediated collagen disruption to predict clinical outcomes in gynecologic malignancies**”. The modifications introduced in this new version have been color-coded to facilitate comparison with the original submission. Specific responses are in dark red italics while modifications to the manuscript are in blue.

We believe we have addressed the comments and have updated our manuscript accordingly. Detailed responses are outlined below. We hope that this new version satisfactorily addresses the concerns. Thank you for your consideration.

Best regards,

Anant Madabhushi, PhD, FAIMBE, FIEEE, FNAI
Robert W. Woodruff Professor
Wallace H. Coulter Department of Biomedical Engineering,
Radiology and Imaging Sciences, Biomedical Informatics, Pathology, Urology.
Georgia Institute of Technology and Emory University
Research Career Scientist
Atlanta Veterans Administration Medical Center
Health Sciences Research Building II
1750 Haygood Drive, Suite N647
Atlanta, Georgia 30322
E-mail: anantm@emory.edu
Youtube Video: <https://tinyurl.com/y987ghao>
Wiki: https://en.wikipedia.org/wiki/Anant_Madabhushi
Twitter: [@anantm](https://twitter.com/anantm)

Reviewer #1 (Remarks to the Author)

“The study of Aggarwal et al. gained clarity during the first revisions. The implementation of pathologist is highly appreciated. However, there are still some minor points to be considered”

- Comment 1: *“Figure 1 would profit from examples of real life heatmap CollaTil images aside the schematic view”*

Response 1: We appreciate the reviewer's input on improving Figure 1 for the paper. We have removed Figure 1 from main manuscript as there is no figure allowed in the Introduction section of the paper.

- Comment 2: *“Figure 8 about cell lines seems a little bit out of topic as the complexitiy of the TME is not reflected by cell lines only. This information could go into supplemental”*

Response 2: We appreciate the reviewer’s comment. We have moved Figure 8 of the main manuscript to the supplementary information file.

- Comment 3: *“Instead of this supplemental figure1 could enter the main manuscript.”*

Response 3: We appreciate the reviewer’s comment. We have moved supplementary figure 1 to the main manuscript.

- Comment 4: *“The same could be discussed with tables. From a personal point of view, feature tables should be with the main text, but the cohort description could enter the supplemental.”*

Response 4: We have moved the feature tables to the main manuscript and the cohort description to the supplementary information file.